# SWE-Flow: Synthesizing Software Engineering Data in a Test-Driven Manner

Lei Zhang[† 1 2]  Jiaxi Yang[1 2]  Min Yang[* 1]  Jian Yang[3]  Mouxiang Chen[4]  Jiajun Zhang[5]  Zeyu Cui[3]
Binyuan Hui[* 3]  Junyang Lin[3]

## Abstract

We introduce **SWE-Flow**, a novel data synthesis framework grounded in Test-Driven Development (TDD). Unlike existing software engineering data that rely on human-submitted issues, **SWE-Flow** automatically infers incremental development steps directly from unit tests, which inherently encapsulate high-level requirements. The core of **SWE-Flow** is the construction of a Runtime Dependency Graph (RDG), which precisely captures function interactions, enabling the generation of a structured, step-by-step *development schedule*. At each step, **SWE-Flow** produces a partial codebase, the corresponding unit tests, and the necessary code modifications, resulting in fully verifiable TDD tasks. With this approach, we generated 16,061 training instances and 2,020 test instances from real-world GitHub projects, creating the **SWE-Flow-Bench** benchmark. Our experiments show that fine-tuning open model on this dataset significantly improves performance in TDD-based coding. To facilitate further research, we release all code, datasets, models, and Docker images at [Github](#).

## 1. Introduction

In recent years, Large Language Models (LLMs) have achieved remarkable performance in code-related tasks (Chen et al., 2021). Training on large-scale code data, these models have made significant advancements in code completion, generation, debugging, and refactoring within software engineering (Rozière et al., 2023; Guo et al., 2024; Hui et al., 2024; Huang et al., 2024). As a result, numerous LLM-powered code applications have emerged, including *GitHub Copilot*[1], which provides code completion and answers to programming-related queries; *Cursor*[2], which enables cross-file code modifications; and *Devin*[3], an autonomous agent designed for fully automated software development. These tools are becoming essential for enhancing developer productivity and advancing intelligent software engineering.

Despite their impressive capabilities, current LLMs still face limitations when applied to real-world software development. Existing evaluations, such as HumanEval (Chen et al., 2021) and MBPP (Austin et al., 2021), primarily assess standalone function implementations, whereas practical development involves complex dependencies, incremental modifications, and multi-file interactions. Constructing datasets and evaluation methodologies that more accurately reflect real-world development challenges is thus a critical and ongoing research problem. Recent efforts, such as SWE-Bench (Jimenez et al., 2023), have attempted to bridge this gap by mining *Github Issues* from open-source projects, capturing authentic bug fixes and feature enhancements. However, this approach heavily depends on the availability and quality of human-submitted issues, requiring extensive data cleaning and filtering. Furthermore, the reliance of SWE-Bench on human-generated commits derived from issue reports fails to encompass the full spectrum of development tasks and variations, thereby overlooking key aspects of the iterative and complex nature of real-world software development.

To address these challenges, we introduce **SWE-Flow**, a reverse data synthesis approach centered on Test-Driven Development (TDD) (Beck, 2002). TDD is a highly structured methodology in which development is driven by test cases: developers write tests first, then implement the required functionality, and finally verify correctness by executing the tests. **SWE-Flow** automatically infers the incremental development process directly from unit tests, thereby generating high-quality training instances. The key insight is

---

[†]Work during an internship at Alibaba Qwen.

[1]Shenzhen Institutes of Advanced Technology, Chinese Academy of Sciences, Shenzhen, China [2]University of the Chinese Academy of Sciences, Beijing, China [3]Alibaba Group, Beijing, China [4]Zhejiang University, Hangzhou, China [5]University of Science and Technology of China, Hefei, China. Correspondence to: Min Yang <min.yang@siat.ac.cn>, Binyuan Hui <binyuan.hby@alibaba-inc.com>.

*Proceedings of the $42^{nd}$ International Conference on Machine Learning*, Vancouver, Canada. PMLR 267, 2025. Copyright 2025 by the author(s).

---

[1]https://github.com/features/copilot
[2]https://www.cursor.com
[3]https://devin.ai

that each unit test inherently represents a high-level expression of requirements. It specifies the behaviors the code must exhibit and implicitly encodes the developer's intention and design considerations. Consequently, **SWE-Flow** eliminates the need for human commit histories by harnessing TDD to automatically produce development tasks with clear structures and explicit goals. Concretely, **SWE-Flow** captures the function call relationships during unit test execution to construct a Runtime Dependency Graph (RDG) for the entire project. This tactic overcomes the limitations of traditional static code analysis tools, which often struggle to accurately parse the dependencies of functions and variables. Drawing on the RDG, **SWE-Flow** generates a project *development schedule* that delineates how an entire codebase can be built from scratch in an incremental manner. At each step, new functions must be implemented on top of existing functionality to pass the corresponding unit tests. For each development step, **SWE-Flow** produces three types of training instances: **(i)** Partial Codebase: The codebase is stripped of the functions that need to be implemented in the current step, simulating the state of incomplete development. **(ii)** Requirement Document: Unit tests associated with the current step provides a high-level specification of the required functionality. **(iii)** Reference Solution (diff): The difference between the complete codebase and the partial codebase, serving as a guide for the development task. **SWE-Flow** offers three major advantages:

- **Verifiability**: All data is centered on unit tests, ensuring generated code is both executable and verifiable.

- **Scalability**: Given any codebase with unit tests, **SWE-Flow** can easily synthesize TDD-compliant training data, obviating the need for excessive data filtering.

- **Configurability**: **SWE-Flow** allows tuning the difficulty level based on the complexity of function calls, providing various levels of LLM training and evaluation.

Using **SWE-Flow**, we synthesized 16,061 training instances and 2,020 test instances from open-source GitHub projects, and we introduce **SWE-Flow-Bench**, a specialized benchmark for evaluating LLM performance in TDD-oriented tasks. Furthermore, we fine-tuned Qwen2.5-Coder-32B-Instruct on data generated by **SWE-Flow**. Experimental results demonstrate that **SWE-Flow** data significantly enhance the TDD development capabilities of the LLM, thus validating its effectiveness.

In summary, our contributions are as follows:

- We propose a novel TDD-based data synthesis strategy that effectively enhances LLM performance in incremental development tasks.

- We present a dedicated benchmark for evaluating LLMs on realistic software engineering tasks, addressing a significant gap in existing assessment methods.

- We generate 16,061 training instances and fine-tune Qwen2.5-Coder-32B-Instruct, demonstrating the efficacy of **SWE-Flow** data in empirical studies. We publicly release all code, models, datasets, and Docker images, fostering further research in the community.

Additionally, data generated by **SWE-Flow** has the potential to support two future directions. Firstly, by scaling the **SWE-Flow** data in pre-training, one can further strengthen LLM capabilities in software engineering tasks. Secondly, since **SWE-Flow** provides verifiable correctness feedback, it can be integrated into reinforcement learning.

## 2. Preliminaries

To ensure clarity and consistency, we define the key notations used in the **SWE-Flow** framework in this section.

### 2.1. Definition of Software Engineering Data

**Definition 2.1. Software Engineering Data (SED).** An SED instance is a tuple $(C, S, G, T)$, where:

- $C$ (**Codebase**): partially implemented software;

- $S$ (**Specification**): textual requirements;

- $G$ (**Ground-Truth Patch**): expected implementation;

- $T$ (**Unit Test**): test cases validating $G$.

Given a dataset $\{(C_i, S_i, G_i, T_i)\}$, we train an LLM $M$ so that $M(C_i, S_i) \to G_i$, with correctness verified by $T_i$.

### 2.2. Definition of Function Node

A **Function Node (FN)** represents a function in a code repository, uniquely identified by the triplet: `filepath`, `lineno`, and `function name`. `filepath` is the relative path of the file from the root of the repository, `lineno` is the starting line number where the function is defined, and `function name` is the name of the function. Based on their roles and calling relationships during unit testing, *Function Nodes* are categorized into four distinct types, as defined below:

**Definition 2.2. Target Test Function Node (TTFN).** A *Target Test Function Node* serves as the entry point for a unit test. It is explicitly invoked by the testing framework to initiate the testing process. For instance, in frameworks like `pytest` or `unittest`, functions prefixed with `test_` are typical *Target Test Function Nodes*.

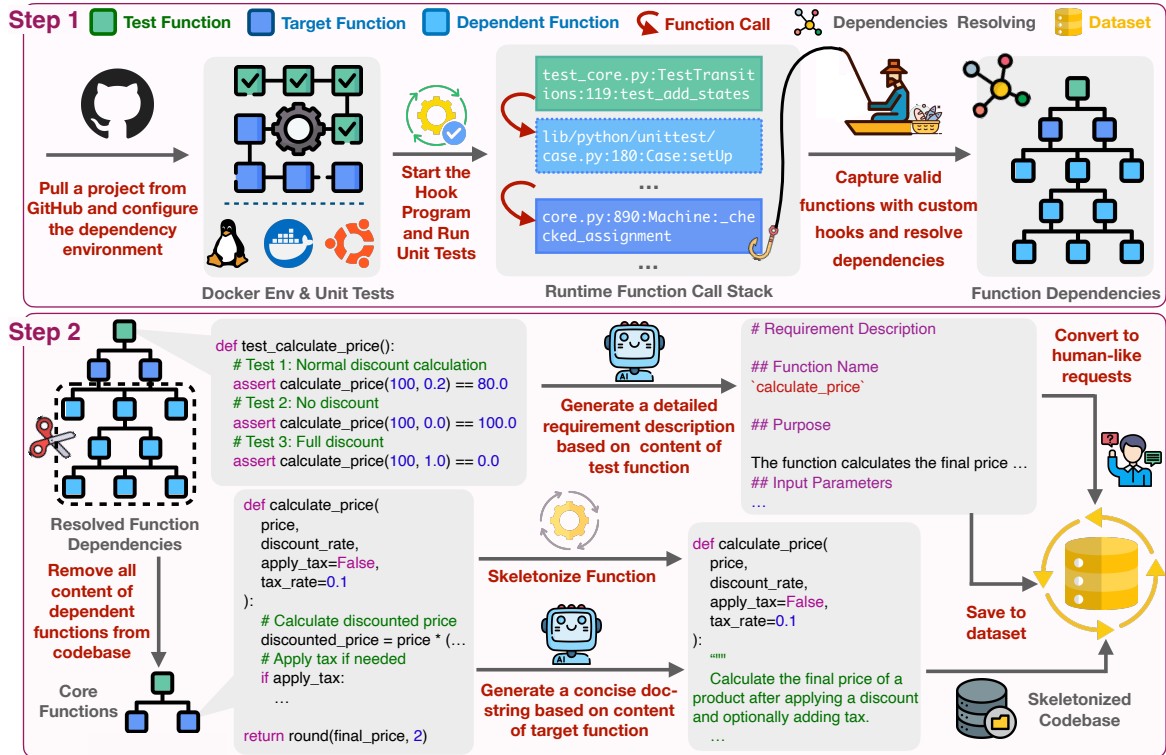

*Figure 1.* The framework of **SWE-Flow**. **Step 1:** Given a codebase and its corresponding development environment, sweflow executes unit tests, constructs the project's Runtime Dependency Graph (RDG), and generates a development schedule. **Step 2:** Based on the development schedule, sweflow removes the implementation of core functions covered by the current step's test functions, forming an incomplete codebase for development. Additionally, it generates a development document based on the content of the test functions.

**Definition 2.3. Dependent Test Function Node (DTFN).** A *Dependent Test Function Node* is designed to assist the *Target Test Function Node* in completing its execution. These nodes typically include setup, teardown, or other functions required for test environment management.

**Definition 2.4. Target Core Function Node (TCFN).** A *Target Core Function Node* represents the core functionality explicitly invoked by a *Target Test Function Node* during the test execution. These nodes are the primary focus of the test and are integral to the functionality being verified.

**Definition 2.5. Dependent Core Function Node (DCFN).** A *Dependent Core Function Node* supports the *Target Core Function Node* by providing auxiliary core functionality required for its implementation. These nodes are indirectly invoked during the execution of the test and are essential for the successful operation of the *Target Core Function Node*.

### 2.3. Definition of Runtime Dependency Graph

**Definition 2.6. Runtime Dependency Graph (RDG).** A *Runtime Dependency Graph* is a directed graph denoted as $G = (V, E)$, where:

- $V$ is the set of nodes, representing the *Function Nodes* that are invoked during execution. Each node $v \in V$ corresponds to a *Function Node* in Section 2.2.

- $E \subseteq V \times V$ is the set of directed edges, where an edge $(u, v) \in E$ indicates the function represented by node $u$ directly calls the function represented by node $v$.

## 3. SWE-Flow

### 3.1. Overview

As illustrated in Figure 1, the **SWE-Flow** framework consists of two main steps: **1.** Given a GitHub project along with its corresponding development environment (e.g., a Docker container), **SWE-Flow** first executes unit tests to build the project's RDG; **2.** Based on the constructed RDG, **SWE-Flow** generates a development schedule for the project and synthesizes software engineering data accordingly. The following sections provide a detailed explanation.

### 3.2. Runtime Dependency Graph Generation

As illustrated in the upper part of Figure 1, given the source code of a GitHub project along with its corresponding development environment (e.g., a Docker container), **SWE-Flow**

**Algorithm 1:** The procedure of **SWE-Flow-Trace**

---

**Input:** Test program `test_program`
**Output:** Function call relationship dictionary
`func_call_relations`
Initialize an empty stack `func_call_stack` and
dictionary `func_call_relations`;
**Procedure** *TraceCalls(frame, event)*:
    Generate a unique function ID from `frame`;
    **if** *event is call and the function ID belongs to the current project* **then**
        Retrieve the caller information from
        `func_call_stack`;
        Push the current function call information onto
        `func_call_stack`;
        Add the caller-callee relationship to
        `func_call_relations` if it is new;
    **else if** *event is return and the function ID matches the top of* `func_call_stack` **then**
        Pop the top entry from `func_call_stack`;

**Procedure** *sweflow-Trace(test_program)*:
    Monitor the function call stack in memory with
    `TraceCalls`;
    Execute the `test_program`;
    Stop monitoring and save the relationships in
    `func_call_relations`;
Execute `sweflow-Trace(test_program)` to
collect the function call relationships;

---

**Algorithm 2:** The procedure of **SWE-Flow-Schedule**

---

**Input:** RDG set $\mathcal{S}_{\text{RDG}}$
**Output:** Development schedule $\mathcal{P}$
Initialize empty list `devSchedule`, map
`funcNodeToTestMap`, and set
`developedFuncs`;
**foreach** *item* $\in \mathcal{S}_{\text{RDG}}$ **do**
    **if** *item.funcNodes* $\in$ *funcNodeToTestMap*
    **then**
        Merge item.*targetTestFuncs* into
        funcNodeToTestMap[item.*funcNodes*];
    **else**
        funcNodeToTestMap[item.*funcNodes*] $\leftarrow$
        item.*targetTestFuncs*;
Sort `funcNodeToTestMap` by the size of
`funcNodes` and store in `sortedMap`;
**foreach** *entry* $\in$ `sortedMap` **do**
    newFuncNodes $\leftarrow$
    entry.*funcNodes* \ developedFuncs;
    **if** *newFuncNodes* $\neq \emptyset$ **then**
        Add `newFuncNodes` to
        `developedFuncs`;
        Append
        {entry.`targetTestFuncs`, `newFuncNodes`}
        to `devSchedule`;
    **else**
        Merge entry.`targetTestFuncs` into the last
        `targetTestFuncs` of `devSchedule`;

**return** *devSchedule*;

---

first collects unit test information across the entire project to identify all target test function nodes. The collected set of test functions is denoted as $\mathcal{F}_{\text{TTFN}}$. Subsequently, **SWE-Flow** employs a customized hook program, **SWE-Flow-Trace**, to execute all unit tests in parallel while recording the function call relationships during each test execution. The following command demonstrates how **SWE-Flow-Trace** executes unit tests in a `Python` project via the terminal:

```
sweflow-trace pytest test_case_id
```

During the execution phase, **SWE-Flow-Trace** continuously monitors the function call stack, automatically collecting all function call relationships generated during runtime. As illustrated in the *Runtime Function Call Stack* of Figure 1, the execution of test functions frequently invokes functions from external sources, such as system libraries or third-party dependencies. To ensure the relevance of the collected data, **SWE-Flow-Trace** filters out these extraneous calls and retains only those associated with the current project repository. The relationships retained through this filtering process naturally form the *Runtime Dependency Graph* (RDG). Once all unit tests have been executed, we obtain a set of TTFNs and their corresponding RDGs, for-

mally:

$$\mathcal{S}_{\text{RDG}} = \{(f, \text{RDG}(f)) \mid f \in \mathcal{F}_{\text{TTFN}}\},$$

where $\text{RDG}(f)$ denotes the *Runtime Dependency Graph* rooted at the *Target Test Function Node* $f$. The algorithm 1 detailed implementation of **SWE-Flow-Trace**.

### 3.3. Development Schedule Generation

The procedure of generating the development schedule is detailed in Algorithm 2. Given the constructed set $\mathcal{S}_{\text{RDG}}$, we first merge all TTFNs that cover the same Core Function Nodes (CFNs). Next, we sort the elements in $\mathcal{S}_{\text{RDG}}$ in ascending order based on the number of CFNs each element covers. The sorted set is denoted as $\mathcal{S}_{\text{RDG}}^{\text{sorted}}$. We then iterate sequentially through $\mathcal{S}_{\text{RDG}}^{\text{sorted}}$, further merging any TTFNs whose CFNs have already been *developed* into the preceding element in the set. This ensures that each step in the generated *Development Schedule* corresponds to a valid incremental development process.

After the iteration, we obtain a development schedule $\mathcal{P}$ that

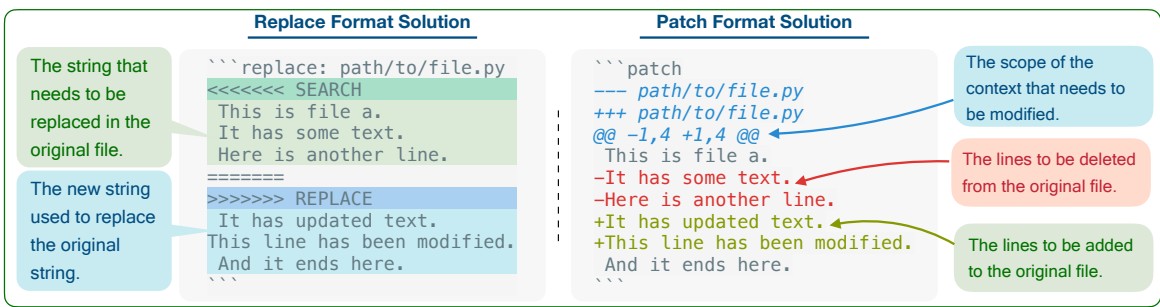

*Figure 2.* Examples of the **Replace Format** and **Patch Format** solution. The **left** side presents an example of a Replace Format solution, which follows the GitHub merge conflict format. The **right** side shows an example of a Patch Format solution, which can be directly generated using diff tools.

naturally satisfies the topological dependency order. The development schedule $\mathcal{P}$ is formally defined as:

$$\mathcal{P} = \{(F_{\text{TTFN}}(i), F_{\text{TCFN}}(i), F_{\text{DCFN}}(i))\}_{i=1}^{N},$$

where $F_{\text{TTFN}}(i)$, $F_{\text{TCFN}}(i)$, and $F_{\text{DCFN}}(i)$ represent the sets of TTFNs, TCFNs, and DCFNs at the $i$-th step of development.

The lower part of Figure 1 illustrates the detailed process of synthesizing software engineering data based on the generated development schedule $\mathcal{P}$. The constructed dataset $\mathcal{D}$ is formally defined as:

$$\mathcal{D} = \{(C_i, S_i, G_i, T_i)\}_{i=1}^{N},$$

where $C_i$, $S_i$, $G_i$ and $T_i$ denote the awaiting codebase, development task, ground-truth solution and corresponding unit tests at the $i$-th step of development, respectively. The detailed construction for each entry in $\mathcal{D}$ is described in the following sections.

### 3.4. Development Document Generation

For each entry in the development schedule $\mathcal{P}$, the content of the target test function nodes (TTFNs) is provided as input to an LLM alongside two-shot examples. The LLM is then tasked with generating a detailed development document, also referred to as a task description, denoted as $S_i$, based on the content of the test function.

To assess the quality of development documents generated by different LLMs from unit test functions, we manually reviewed a sample of documents produced by state-of-the-art LLMs, including gpt-4o (Hurst et al., 2024), claude-3.5-sonnet (Anthropic, 2024), DeepSeek-V3 (Liu et al., 2024a), and Qwen2.5-Coder-32B-Instruct (Hui et al., 2024). Our results indicate that, under a two-shot setting, the quality of the generated requirement documents remains largely consistent across these models. Given this finding, we opted to use the Qwen2.5-Coder-32B-Instruct for development document generation due to its accessibility and cost-effectiveness.

Representative examples of development documents generated by the Qwen2.5-Coder-32B-Instruct are provided in the Appendix C.

This approach leverages the structured and goal-driven nature of unit tests to ensure the generated development tasks are well-aligned with specifications. The rationale for this method is grounded in several key considerations. Firstly, **Test-Driven Development (TDD) Philosophy.** Modern software methodologies, such as TDD (Beck, 2002), advocate writing tests prior to implementation, treating tests as documentation of intended behaviors. Generating development tasks from test functions aligns seamlessly with this by tracing requirements directly to these pre-defined specifications. Secondly, **Explicit Functional Context.** Unit tests act as executable specifications, defining precise software behavior through concrete inputs, expected outputs, and assertions. This structured context enables us to derive implementations directly from test cases, minimizing ambiguity. Thirdly, **Precision and Minimal Ambiguity.** Unit tests enforce strict, verifiable constraints, eliminating ambiguity inherent in natural language requirements. Their deterministic validation ensures alignment with functional expectations, preventing unintended deviations.

### 3.5. Development Codebase Generation

Given an entry from the development schedule $\mathcal{P}$, the codebase generation process begins with the original codebase and involves systematically reducing its content through a process known as skeletonization. For *Target Core Function Nodes* and *Dependent Core Function Nodes*, we employ distinct skeletonization strategies to modify the codebase.

**Target Core Function Nodes.** The concrete implementations of functions in the original codebase are entirely removed. To retain contextual information, we utilize an LLM to generate a new docstring based on the original function content. This new docstring either replaces the existing one or supplements it if none was originally present.

**Dependent Core Function Nodes.** Both the function definitions and their contents are completely removed from the codebase. This step ensures flexibility for the subsequent development process, allowing developers to build freely without constraints imposed by the original framework.

The skeletonized codebase $C_i$ is then used to develop the corresponding development task.

### 3.6. Ground-Truth Solution Generation

The differences between the original codebase and the skeletonized codebase naturally form a ground-truth solution. In real-world software development, generating the complete content of edited files can be both time-consuming and cost-prohibitive. Therefore, following the settings of existing evaluations, we define two formats for representing the ground-truth solution: **Replace Format** and **Patch Format**. These two formats minimize overhead by focusing on localized changes rather than reconstructing entire files. Specific examples of each format are provided in Figure 2.

### 3.7. Dataset Construction

To construct a high-quality dataset for enhancing LLMs in development tasks, we followed a systematic process involving the selection, preparation, and processing of active `Python` projects. The key steps are detailed below.

**Collection of Projects.** We selected 150 of the most popular and actively maintained Python projects from the libraries.io[4] platform. Each project in our dataset meets the following criteria: **1.** It has a recognized open-source license (e.g., MIT, Apache-2.0). **2.** It has received at least 2,000 stars. **3.** It has demonstrated recent activity, with updates made within the past six months. For each selected project, we cloned the latest codebase from GitHub and recorded the corresponding commit hash to ensure reproducibility.

**Preparation of Test Environment.** To ensure a consistent and isolated environment for processing each project, we created a custom script for installing project dependencies within a Docker container with following steps: **1.** Launches a Docker container to provide an isolated environment. **2.** Installs basic or optional dependencies specified in the project's `pyproject.toml` or `setup.py` files. **3.** Iteratively parses and installs dependencies listed in any `requirements.txt`-formatted files within the project. This step ensures that the project is fully prepared for subsequent analysis. Finally, we obtain 74 projects that can pass all the unit tests in the installed test environment. Among these projects, 12 projects are selected for testing, and the remaining 62 projects are used for training.

**Verifiable Data Generation.** After setting up the project environment, we used **SWE-Flow** to collect comprehensive unit test information for each project. We then executed these unit tests in parallel using **SWE-Flow-Trace**, recording detailed function call information associated with each unit test. We only keep the unit tests that pass. The output of this step is a Runtime Dependency Graph (RDG) for each unit test, capturing the functional dependencies and relationships within the project.

Utilizing the RDGs generated in the previous step, we employed **SWE-Flow-Schedule** to produce a development plan for each project. Based on the development schedule, we performed skeletonization of the original codebase, a process that simplifies the codebase while retaining its structure. To construct the ground-truth solutions: First, we compared the skeletonized codebase with the original codebase using a diff tool to produce **Patch Format** ground-truth solutions. Second, we further converted these patch-format solutions into **Replace Format** ground-truth solutions for additional versatility in downstream tasks.

Following this process, we synthesized a comprehensive dataset that includes 16,061 training instances and 2,020 test instances, tailored to improve and evaluate the performance of AI systems in real-world software development scenarios.

## 4. SWE-Flow-Bench

### 4.1. Evaluation Framework

**Language Model Evaluation.** For the evaluation of language models, we first construct a task prompt, which consists of two main parts: the system prompt and the user prompt. In the *system prompt*, we define the requirements for the language model to act as an experienced software engineer, specifying the basic expectations for completing the software development task. Additionally, we clarify the solution format and provide specific examples to ensure consistency. In the *user prompt*, we include relevant files from the codebase that are directly related to the current development task, followed by a detailed description of the task requirements. Finally, we reiterate the instructions for the solution format to maintain alignment and clarity. Figure 5 in the appendix provides an example of the task prompt.

We then prompt the language model to generate a solution for the software development task based on the constructed prompt. The response generated by the language model is parsed to extract the solution that conforms to the specified format. This extracted solution is applied to the corresponding codebase under development. Finally, unit tests are executed on the updated codebase to verify whether the task has been successfully completed.

---

[4]https://libraries.io

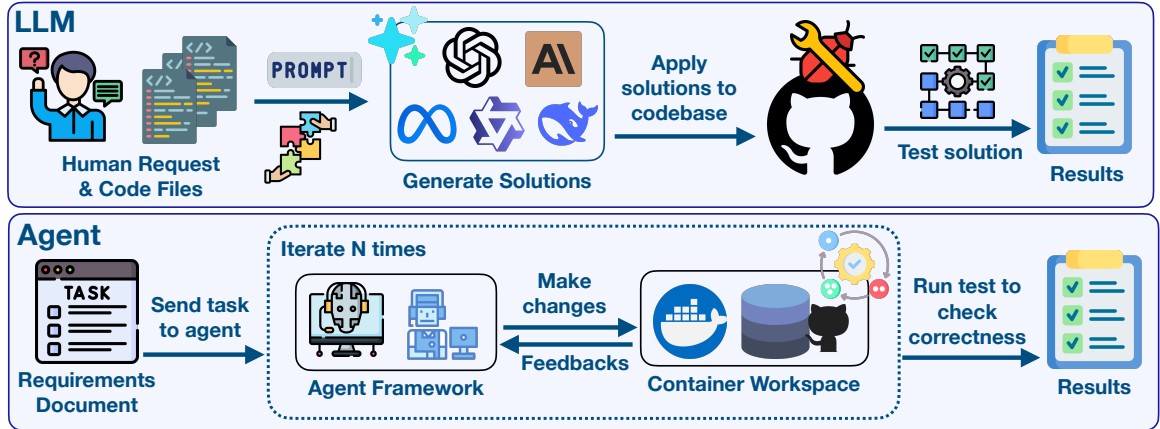

*Figure 3.* The evaluation framework of **SWE-Flow-Bench**. **Upper:** A prompt containing the current codebase information, development document, and output format is sent to the LLM. The LLM generates a response based on the prompt's requirements. A post-processing tool then extracts the solution from the LLM's response, applies it to the codebase, and executes the corresponding unit tests to verify correctness. **Lower:** Given an incomplete codebase and a development document, an agent iteratively performs development until the task is completed or a preset iteration limit is reached. After the agent terminates, the corresponding unit tests are executed to assess the correctness of the development.

**Agent Evaluation.** For the evaluation of agents, we mount the codebase under development to the agent's workspace directory and send the task requirement document from the test samples as the task to be completed. The agent then begins iteratively working on the development task. Complex software engineering tasks often require numerous iterations to be completed, and in some cases, the development may remain incomplete even after multiple iterations. To minimize testing overhead, we set a maximum number of iterations. If the agent exceeds this limit but continues running normally, we manually terminate the task execution. After the agent's task execution is terminated, we run the unit tests corresponding to the task to verify whether the development task has been successfully completed.

### 4.2. Evaluation Metrics

**Pass Rate.** Execution pass rate is the most direct metric to assess the correctness of the generated code. Following previous research on code evaluation (Chen et al., 2021), we adopt the pass rate as the primary metric to evaluate the performance of language models and agents. The pass rate is defined as the ratio of successfully completed tasks (those that pass unit tests) to the total number of development tasks. The pass rate is calculated as follows:

$$\text{Pass Rate} = \frac{\sum_{i=1}^{\#\text{ tasks}} \text{isPass}(\text{task}_i)}{\#\text{ tasks}},$$

where $\text{isPass}(\text{task}_i)$ is a binary variable indicating whether the $i$-th task passes unit tests.

**Efficiency Value.** In addition, for the evaluation of agents, we also use the efficiency value as a metric. The efficiency value measures the efficiency of completing a given development task. Its formal definition is as follows:

$$\text{Efficiency Value} = \frac{\text{Pass Rate}}{\log(\sum_{i=1}^{\#\text{ tasks}} \text{Iters}(\text{task}_i))},$$

where $\text{Iters}(\text{task}_i)$ is the number of iterations the agent used to complete the $i$-th task or the max number of iterations set by the user if the agent exceeds the max number of iterations.

## 5. Experiments

### 5.1. Fine-tuning Language Models

To validate the effectiveness of the synthesized training data, we fine-tuned the Qwen2.5-Coder-32B-Instruct model (Hui et al., 2024) using the generated dataset. The entire training process was completed within two hours on 128 H800 GPUs using Megatron-LM (Shoeybi et al., 2019). For a detailed description of the training process and parameters, please refer to Appendix D.

### 5.2. Experimental Results

**Large Language Models.** We conducted a comprehensive evaluation of 11 mainstream LLMs on the **SWE-Flow-Bench (Lite)** benchmark. Furthermore, we compared their performance with our UF-Coder-32B-Instruct, which was fine-tuned from Qwen2.5-Coder-32B-Instruct (Hui et al., 2024) using our synthesized training data. Figure 4 illustrates the ability of these LLMs to generate solutions in both **Replace** and **Patch** formats. For more detailed evaluation results, please refer to Table 5 in the Appendix.

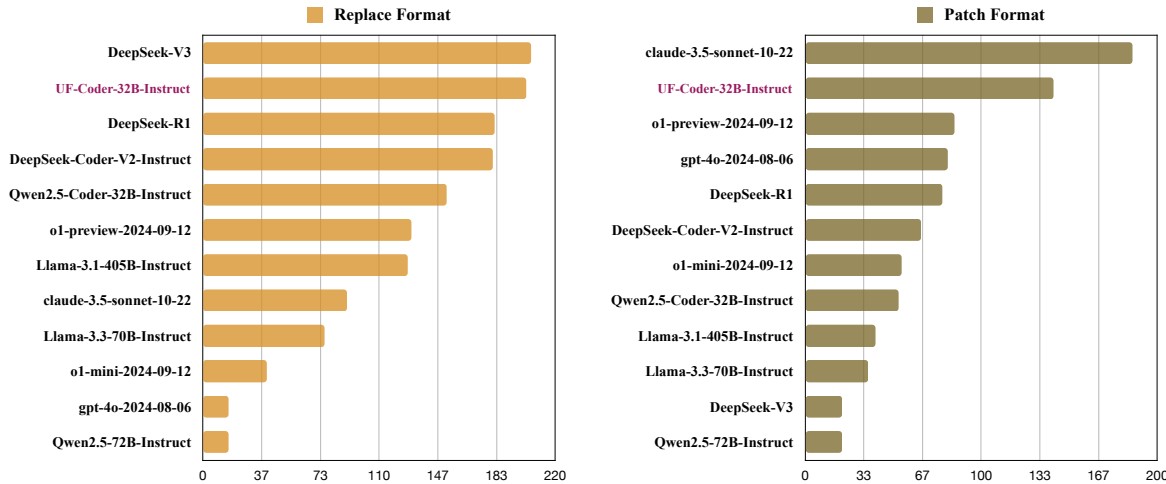

*Figure 4.* The overview of evaluation results of large language models on SWE-Flow-Bench (Lite). The x-axis represents the number of development tasks for which the solutions generated by LLMs successfully pass unit tests.

As shown in Figure 4, the **SF-Coder-32B-Instruct** model achieved a significant performance improvement over the Qwen2.5-Coder-32B-Instruct. Specifically, its performance in the **Replace Format** generation was second only to that of DeepSeek-V3 (Liu et al., 2024a), while in the **Patch Format** generation, it ranked just below claude3.5-sonnet (Anthropic, 2024), outperforming all other models by a substantial margin. The experimental results demonstrate that the training data synthesized using the **SWE-Flow** framework can significantly enhance the code generation capabilities of LLMs in real-world software development scenarios.

**Agents.** **SWE-Flow-Bench** can be integrated with any Agent framework for evaluation. In this study, we selected OpenHands (Wang et al., 2024a), the most widely used Code Agent framework, for testing. Specifically, we evaluated claude-3.5-sonnet, gpt-4o, deepseek-chat, and Qwen2.5-Coder-32B-Instruct on the **SWE-Flow-Bench (Lite)** test set. The evaluation results are presented in Figure 5 and Table 6 in the appendix.

The experimental results show that claude3.5-sonnet (Anthropic, 2024) significantly outperformed other models, including gpt-4o (Hurst et al., 2024), in real-world software development scenarios. However, even claude3.5-sonnet struggled to complete those complex development tasks, highlighting the substantial limitations of current large language models in handling practical software engineering challenges. This observation underscores the pressing need for further advancements in the field. Specifically, there is a clear requirement for more comprehensive and domain-specific software development datasets to enhance the training of large language models. Such improvements would be essential to bridge the gap between their current capabilities and the demands of real-world software development tasks.

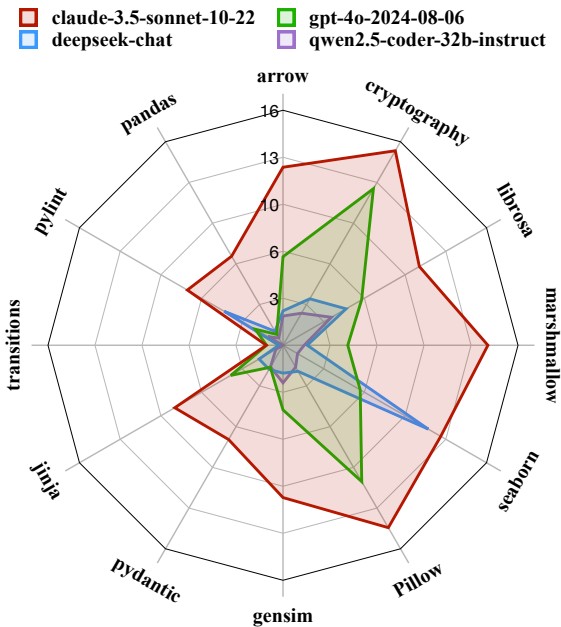

*Figure 5.* Efficiency values of various LLMs integrated with Open-Hands. Each axis represents a specific software engineering task, with values indicating the corresponding efficiency scores of the agent for that task.

## 6. Discussion and Future Work

**1.** *Synthesizing More Challenging Data*: By merging consecutive tasks from **SWE-Flow-Schedule**, we can create more complex development scenarios. In the extreme case, combining all tasks into one forces an entire project to be built from scratch. **2.** *Enhancing Reinforcement Learning*: Recent reasoning-aware models need large-scale, verifiable data. **SWE-Flow**-generated tasks are inherently testable in containerized environments, making them ideal for training

*Table 1.* Comparison of existing software engineering benchmarks.

| Dataset | Automated Synthesis | TDD Based | Configurable Tasks | Cross-File Editing | Complex Dependencies |
|---|---|---|---|---|---|
| SWE-Bench | ✗ | ✗ | ✗ | ✓ | ✓ |
| Commit0-Bench | ✗ | ✗ | ✗ | ✓ | ✓ |
| **SWE-Flow-Bench** | ✓ | ✓ | ✓ | ✓ | ✓ |

RL-based code models. **3.** *Enhancing Pre-training*: Pre-training data determines an LLM's core capabilities. Current open-source models lack sufficient high-quality, verifiable software engineering data. By leveraging CI-enabled GitHub projects, **SWE-Flow** can synthesize large-scale verified corpora, potentially boosting LLM performance in code generation and software development.

## 7. Related Work

SWE-Bench (Jimenez et al., 2023) evaluates agents on GitHub issue resolution, emphasizing patch-level fixes rather than broader development capabilities. Commit0-Bench (Zhao et al., 2024) requires models to generate full implementations in one attempt using all unit tests, which does not align with real-world iterative coding practices. In contrast, **SWE-Flow-Bench** follows a Test-Driven Development approach, breaking projects into incremental steps guided by minimal test cases. This enables a finer-grained evaluation of code organization, architecture construction, and functionality expansion while improving interpretability and real-world relevance. For more discussion of related work, see Appendix A.

## 8. Conclusion

We propose the **SWE-Flow** framework for generating verifiable software engineering data, along with the **SWE-Flow-Bench** framework for evaluating the performance of large language models (LLMs) and AI agents on real-world software development tasks. Using synthetic software engineering data, we fine-tuned the Qwen2.5-Coder-32B-Instruct, resulting in the **SF-Coder-32B-Instruct** model, which demonstrated significant performance improvements on **SWE-Flow-Bench (Lite)**, thereby validating the effectiveness of **SWE-Flow**-synthesized data. Moreover, the synthetic data generated by **SWE-Flow** holds potential for pre-training and post-training, further enhancing the AI coding applications.

## Acknowledgments

Min Yang is supported by National Key Research and Development Program of China (2022YFF0902100), National Natural Science Foundation of China (62376262), the Natural Science Foundation of Guangdong Province of China (2024A1515030166, 2025B1515020032), Shenzhen Science and Technology Innovation Program (KQTD20190929172835662).

## Impact Statement

This study introduces the **SWE-Flow** framework, designed to synthesize test-driven, execution-verified software engineering data to enhance the software engineering capabilities of LLMs. Additionally, we propose **SWE-Flow-Bench**, a benchmarking framework for evaluating the performance of LLMs and LLM-based agents on software engineering tasks. All datasets and models used in this study are open-source and comply with their respective licenses. We hope that our findings and contributions will facilitate future research in this field and further advance AI technologies, particularly in the domain of software engineering.

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

# A. Additional Related Work

## A.1. Code Large Language Models

The advent of large language models (LLMs) tailored for code-centric tasks, such as CodeLlama (Rozière et al., 2023), DeepSeek-Coder (Guo et al., 2024), OpenCoder (Huang et al., 2024), and Qwen2.5-Coder (Hui et al., 2024), has revolutionized software engineering by automating repetitive tasks, proposing code improvements, and facilitating natural language-to-code conversion. These models, trained on vast corpora of billions of code snippets, have significantly enhanced the development process. Notable contributions include Starcoder (Li et al., 2023; Lozhkov et al., 2024), CodeLlama (Rozière et al., 2023), each advancing coding assistance tools with unique innovations. Inspired by the success of grammar-based parsed trees in various domains, we leverage the abstract syntax tree to augment code completion training, further promising greater efficiency and intuitiveness in software creation.

## A.2. Code Agents

Recent research highlights the pivotal role of large language models (LLMs) in the development of AI agents, showcasing their capability in facilitating complex task execution through tool utilization (Schick et al., 2023; Talebirad & Nadiri, 2023; Hong et al., 2023). Notable examples include ToolFormer, which enables tools to be used more effectively by LLMs; Meta-GPT and BabyAGI, which demonstrate advancements in autonomous task management. Studies on self-edit and self-debug have further illustrated the capacity of code models to engage in multi-round interactions for code correction and improvement. Contemporary work also underscores the efficacy of agent systems like OpenDevin (Wang et al., 2024b) and SWE-Agent (Yang et al., 2024b) in handling complex programming tasks at the repository level, such as SWE-Bench (Jimenez et al., 2023; Pan et al., 2024)

## A.3. Code Instruction Tuning with Synthetic Data

Instruction tuning represents a significant advancement in the field of large language models (LLMs) by refining these models with specifically designed instruction datasets, thereby improving their ability to follow instructions more accurately and generalize better (Ouyang et al., 2022b; Zhang et al., 2023b; Wang et al., 2023). This method involves using a foundational LLM to generate initial instruction data, which is then used to fine-tune the model, enhancing its performance through synthetic data (Wang et al., 2023; Chaudhary, 2023; Yang et al., 2024c). To further this approach, WizardCoder (Luo et al., 2023) introduced code Evol-Instruct, utilizing heuristic prompts to increase the complexity and diversity of the synthetic dataset, thus producing higher-quality data. More recently, initiatives such as OSS-Instruct (Wei et al., 2023) and CodeOcean (Yu et al., 2023) have leveraged real-world code snippets to guide LLMs in generating more controllable and realistic instruction corpora.

## A.4. Code Benchmarks

Code edit and generation is a basic task for code language models (LLMs), requiring them to interpret natural language descriptions and generate corresponding code snippets that fulfill user requirements (Gu et al., 2024; Lai et al., 2022; Liu et al., 2023; Yu et al., 2024; Li et al., 2024). To thoroughly evaluate the diverse capabilities of LLMs, numerous benchmarks have been proposed, including code translation (Yan et al., 2023), code retrieval (Huang et al., 2021; Husain et al., 2019; Lu et al., 2021), code completion (Bavarian et al., 2022; Liu et al., 2024b; Zhang et al., 2023a; Yang et al., 2024d), code debugging (Huq et al., 2022; Tian et al., 2024; Liu et al., 2024d), and structured data understanding (Wu et al., 2024; Su et al., 2024). Further, multilingual benchmarks like MultiPl-E, McEval, and MdEval (Cassano et al., 2023; Chai et al., 2024; Liu et al., 2024c) have been proposed to evaluate the multilingual capabilities of code LLMs, ensuring their effectiveness across various languages and applications. Recent studies explore more diverse scenarios (Jain et al., 2024; Zhang et al., 2024; Cheng et al., 2024; Zhuo et al., 2024) to evaluate the model performance across a variety of real-world coding scenarios, such as LiveCodeBench and NaturalCodeBench.

# B. SWE-Flow-Bench

*Table 2.* Information of projects used in SWE-Flow-Bench.

| Project | Functionality | Stars | License | Last Commit | Commit Hash |
|---------|---------------|-------|---------|-------------|-------------|
| arrow | Date and Time | 8.8k | Apache-2.0 | 2024.11.20 | 1d70d00 |
| cryptography | Cryptography | 6.8k | Apache-2.0 & BSD-3-Clause | 2025.01.06 | 4a31be3 |
| gensim | Language Processing | 15.8k | LGPL-2.1 | 2024.12.05 | 8b6b69c |
| jinja | Templating Engine | 10.5k | BSD-3-Clause | 2024.12.22 | 6aeab5d |
| librosa | Audio Processing | 7.3k | ISC | 2024.11.27 | 24270be |
| marshmallow | Serialization | 7.1k | MIT | 2025.11.06 | 71ab95a |
| pandas | Data Analysis | 44.2k | BSD-3-Clause | 2025.01.04 | 8fbe6ac |
| Pillow | Image Processing | 12.4k | MIT-CMU | 2025.01.04 | dfb368a |
| pydantic | Data Validation | 21.9k | MIT | 2025.01.03 | 59b35de |
| pylint | Code Analysis | 5.4k | GPL-2.0 | 2025.01.04 | c21276f |
| seaborn | Data Visualization | 12.7k | BSD-3-Clause | 2024.12.16 | 8fc4051 |
| transitions | Design Patterns | 5.9k | MIT | 2024.08.13 | 4d8d103 |

In this section, we present **SWE-Flow-Bench**, the dataset used in the **SWE-Flow-Bench** benchmarking framework.

## B.1. Statistics of SWE-Flow-Bench

*Table 3.* Statistics of SWE-Flow-Bench.

| Difficulty | Project | Functionality | # Steps | | # Files | | # Functions | | # Context Tokens | | # Patch Tokens | | Dep. Depth | |
|------------|---------|---------------|---------|------|---------|------|-------------|------|------------------|--------|----------------|-------|------------|------|
| | | | Full | Lite | Full | Lite | Full | Lite | Full | Lite | Full | Lite | Full | Lite |
| Easy | arrow | Date and Time | 124 | 50 | 1.00 | 1.00 | 1.1 | 1.0 | 20,633 | 21,728 | 392 | 306 | 1.6 | 1.0 |
| | cryptography | Cryptography | 474 | 50 | 1.00 | 1.00 | 1.4 | 1.0 | 8,130 | 3,951 | 274 | 303 | 2.2 | 1.0 |
| Medium | librosa | Audio Processing | 90 | 50 | 1.01 | 1.00 | 2.1 | 1.2 | 12,642 | 12,727 | 1,832 | 1,274 | 2.3 | 1.4 |
| | marshmallow | Serialization | 70 | 50 | 1.01 | 1.02 | 2.1 | 1.8 | 8,160 | 6,546 | 454 | 363 | 2.8 | 1.8 |
| | seaborn | Data Visualization | 235 | 50 | 1.00 | 1.00 | 2.1 | 1.1 | 9,703 | 8,425 | 846 | 753 | 3.5 | 1.1 |
| | Pillow | Image Processing | 259 | 50 | 1.02 | 1.00 | 2.3 | 1.2 | 11,261 | 8,017 | 627 | 430 | 4.2 | 1.3 |
| Hard | gensim | Language Processing | 223 | 50 | 1.06 | 1.08 | 3.0 | 1.5 | 9,154 | 9,359 | 1,032 | 626 | 4.1 | 1.6 |
| | pydantic | Data Validation | 160 | 50 | 1.01 | 1.00 | 3.0 | 1.5 | 8,087 | 8,018 | 589 | 525 | 6.4 | 1.6 |
| | jinja | Templating Engine | 68 | 50 | 1.09 | 1.10 | 4.2 | 4.7 | 7,234 | 7,243 | 642 | 698 | 9.4 | 5.3 |
| | transitions | Design Patterns | 55 | 50 | 1.02 | 1.02 | 4.6 | 5.0 | 7,761 | 7,794 | 906 | 944 | 8.3 | 7.9 |
| | pylint | Code Analysis | 39 | 39 | 1.05 | 1.05 | 5.1 | 5.1 | 2,376 | 23,76 | 581 | 581 | 4.6 | 4.6 |
| | pandas | Data Analysis | 223 | 50 | 1.17 | 1.16 | 5.3 | 3.8 | 27,076 | 11,663 | 1090 | 813 | 8.0 | 2.8 |

**SWE-Flow-Bench** consists of 2,020 development tasks spanning 12 most popular `Python` software engineering projects, covering a diverse range of software engineering domains, including: **Date and Time**, **Cryptography**, **Language Processing**, **Templating Engines**, **Audio Processing**, **Serialization**, **Data Analysis**, **Image Processing**, **Data Validation**, **Code Analysis**, **Data Visualization**, **Design Patterns**. Table 3 provides detailed statistics on these 12 software engineering tasks, and the metrics in the table are defined as follows:

- **# Steps:** The total number of development steps required to complete the project.

- **# Files:** The average number of files modified per development step.

- **# Functions:** The average number of functions that need to be implemented per development step.

- **# Context Tokens:** The average number of tokens in the contextual code files relevant to each development step.

- **# Patch Tokens:** The average number of tokens in the solution (patch) for each development step.

- **Dep. Depth:** The average dependency depth, representing the number of function calls a development step relies on.

We categorize the 12 software engineering projects into three difficulty levels,easy, medium, and hard, based on the average number of functions that need to be implemented per development step:

- **Easy:** Projects where the average number of functions per step is less than 2.

- **Medium:** Projects where the average number of functions per step is between 2 and 3.

- **Hard:** Projects where the average number of functions per step is greater than 3.

To facilitate efficient validation, **SWE-Flow-Bench** is divided into two splits: *Full* and *Lite*.

- The Full split includes all 2,020 development tasks.

- The Lite split contains only the first 50 development steps from each software project (or all available steps if a project has fewer than 50 development steps), resulting in a total of 589 development tasks.

Since earlier steps in the development process tend to have shallower dependency depths, the Lite split presents a lower level of difficulty compared to the Full split.

### B.2. Comparison Between SWE-Flow-Bench and Existing Software Engineering Benchmarks

Table 1 presents a comparison between **SWE-Flow-Bench** and existing software engineering benchmarks (Jimenez et al., 2023; Zhao et al., 2024), highlighting their similarities and differences.

SWE-Bench (Jimenez et al., 2023) primarily evaluates an agent's ability to resolve GitHub issues, focusing on patch-level code fixes rather than the entire development process. As a result, it cannot systematically assess a code generation model's capabilities in architectural design, functionality expansion, and incremental development.

Commit0-Bench (Zhao et al., 2024), on the other hand, requires models to generate a complete implementation in a single attempt based on all available unit tests. This approach does not align with real-world software engineering practices, as it fails to measure a model's step-by-step code construction performance. Additionally, it lacks interpretability, making it difficult to analyze failure cases.

In contrast, **SWE-Flow-Bench** adopts a Test-Driven Development (TDD) approach, decomposing projects into multiple incremental steps. Each step guides code generation through minimal test cases, enabling a more fine-grained evaluation of a model's ability in code organization, architecture construction, and functionality expansion. Moreover, **SWE-Flow-Bench** better reflects real-world development workflows while providing greater interpretability, allowing for a deeper analysis of model failures.

Overall, **SWE-Flow-Bench** offers a more rigorous and systematic evaluation of code generation models, providing more precise insights for model optimization.

# C. Content of SWE-Flow-synthesized Data

## C.1. Large Language Model Genearted Development Document

We conducted a comparative analysis of gpt-4o (Hurst et al., 2024), claude-3.5-sonnet (Anthropic, 2024), DeepSeek-V3 (Liu et al., 2024a), and Qwen2.5-Coder-32B-Instruct (Hui et al., 2024) in terms of the quality of requirement documents generated based on a given unit test function. Our findings indicate that, under a two-shot setting, the quality of the generated requirement documents remains largely consistent across these models. Given this observation, we selected Qwen2.5-Coder-32B-Instruct, an open-source model, for requirement document generation due to its accessibility and cost-effectiveness. The upper section of Figure 6 presents the content of a target test function from the tiktoken project, while the lower section displays the requirement specification document generated by Qwen2.5-Coder-32B-Instruct based on the test function.

---

**Content of Target Test Function**

```python
def test_basic_encode():
    enc = tiktoken.get_encoding('p50k_base')
    assert enc.encode('hello world') == [31373, 995]
    enc = tiktoken.get_encoding('cl100k_base')
    assert enc.encode('hello world') == [15339, 1917]
    assert enc.encode(' \x850') == [220, 126, 227, 15]
```

**LLM Generated Development Document**

## **Development Requirements**

### **Goal 1: Provide Encoding Functionality for Multiple Models**

#### **Feature Description**
The system should provide encoding functionality that supports multiple tokenization models, including 'p50k_base', and 'cl100k_base'.

#### **Required Functionality**
1. **Encoding with 'p50k_base' Model**
   - Implement functionality to encode the string 'hello world' using the 'p50k_base' model and ensure the output matches [31373, 995].

2. **Encoding with 'cl100k_base' Model**
   - Implement functionality to encode the string 'hello world' using the 'cl100k_base' model and ensure the output matches [15339, 1917].
   - Implement functionality to encode the string ' \x850' using the 'cl100k_base' model and ensure the output matches [220, 126, 227, 15].

## **Methods and Functionality Overview**

| Method Name | Input Parameters | Output/Functionality |
| ----------------------- | ---------------- | ---------------------------------------------------------- |
| tiktoken.get_encoding | model_name: str | Return encoding model instance for the specified model name |
| encode | text: str | Return list of encoded token IDs based on the text and the model instance |

## **Expected Outcomes**

1. The 'p50k_base' encoding model should produce [31373, 995] when encoding the string 'hello world'.
2. The 'cl100k_base' encoding model should produce [15339, 1917] when encoding the string 'hello world'.
3. The 'cl100k_base' encoding model should produce [220, 126, 227, 15] when encoding the string ' \x850'.

---

*Figure 6.* Content of the target test function and the requirement document generated by Qwen2.5-Coder-32B-Instruct based on it.

## C.2. Large Language Model Genearted Doc-string

For doc-string generation, we adopt the same strategy as used for development document generation. Specifically, we employ a 2-shot setting, prompting the Qwen2.5-Coder-32B-Instruct model to generate the corresponding doc-string based on the content of the Core Function. The upper part of Figure 7 presents the content of a Core Function, while the lower part displays the doc-string generated by Qwen2.5-Coder-32B-Instruct. This doc-string will be utilized for code skeletonization.

---

**Content of Target Core Function**

```python
@classmethod
def _iter_from_urlpaths(cls, urlpaths, download_config=None) -> Generator[str, None, None]:
    if not isinstance(urlpaths, list):
        urlpaths = [urlpaths]
    for urlpath in urlpaths:
        if xisfile(urlpath, download_config=download_config):
            yisdirrlpath
        elif xisdir(urlpath, download_config=download_config):
            for dirpath, dirnames, filenames in xwalk(urlpath, download_config=download_config):
                dirnames[:] = sorted([dirname for dirname in dirnames if not
dirname.startswith(('.', '__'))])
                if basename(dirpath).startswith(('.', '__')):
                    continue
                for filename in sorted(filenames):
                    if filename.startswith(('.', '__')):
                        continue
                    yield xjoin(dirpath, filename)
        else:
            raise FileNotFoundError(urlpath)
```

---

**LLM Generated Doc-string**

```
"""
Generates file paths from given URL paths, downloading and iterating through directories as necessary.

Args:
    urlpaths (Union[str, List[str]]): A single URL path or a list of URL paths to iterate over.
    download_config (DownloadConfig, optional): Configuration for downloading files. Defaults to None.

Yields:
    str: A file path from the URL paths.

Raises:
    FileNotFoundError: If a URL path is neither a file nor a directory.
"""
```

*Figure 7.* Content of the target core function and the doc-string generated by Qwen2.5-Coder-32B-Instruct based on it.

---

**Synthesized Reference Patch**

```diff
--- src/datasets/features/features.py
+++ src/datasets/features/features.py
@@ -954,7 +954,21 @@
         bool: `True` if the object and its first elements (if it is a sequence) are not `None`
or empty,
                 otherwise `False`.
    """
-    ...
+    if obj is None:
+        return False
+    elif isinstance(obj, (list, tuple)) and (schema is None or isinstance(schema, (list, tuple,
LargeList, Sequence))):
+        if len(obj) > 0:
+            if schema is None:
+                pass
+            elif isinstance(schema, (list, tuple)):
+                schema = schema[0]
+            else:
+                schema = schema.feature
+            return _check_non_null_non_empty_recursive(obj[0], schema)
+        else:
+            return False
+    else:
+        return True

 def get_nested_type(schema: FeatureType) -> pa.DataType:
    """
```

*Figure 8.* An example of a reference patch in synthetic data from sweflow.

# D. Fine-tuning Language Models

## D.1. Training Dataset

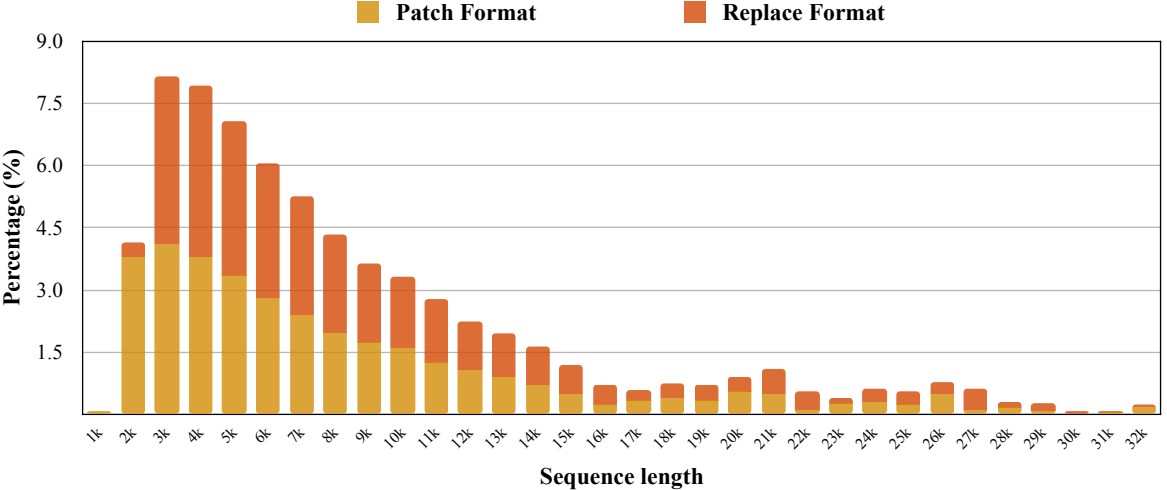

*Figure 9.* The distribution of total tokens

In Section 3.7, we described the process of **SWE-Flow**-synthesized training data generation. Here, we utilize the synthesized dataset to fine-tune the Qwen2.5-Coder-32B-Instruct (Hui et al., 2024) model. The training dataset consists of two data formats: **Patch** and **Replace**. To ensure efficient training, we filter out all samples exceeding 32k tokens. The final sequence length distribution of the training dataset is illustrated in Figure 9.

## D.2. Training Parameters

*Table 4.* Fine-tuning parameters.

| Parameter | Max Seq-len | Batch Size | Training Steps | Warmup Steps | Learning Rate | Min LR | LR Decay |
|---|---|---|---|---|---|---|---|
| **Value** | 32,768 | 1024 | 32 | 6 | 7e-6 | 7e-7 | Linear |

For the training framework, we employ Megatron-LM (Shoeybi et al., 2019), with detailed training parameters provided in Table D.2. The entire training process can be completed within two hours using 128 H800 GPUs.

# E. Detailed Evaluation Results

## E.1. Evaluation Results of Large Language Models

*Table 5.* Evaluation results of language models on **SWE-Flow-Bench (Lite)**.

| Model | arrow | | cryptography | | librosa | | marshmallow | | seaborn | | Pillow | |
|---|---|---|---|---|---|---|---|---|---|---|---|---|
| | replace | patch | replace | patch | replace | patch | replace | patch | replace | patch | replace | patch |
| o1-preview-2024-09-12 | 40.0% | 32.0% | 54.0% | 22.0% | 24.0% | 24.0% | 36.0% | 18.0% | 26.0% | 16.0% | 36.0% | 20.0% |
| o1-mini-2024-09-12 | 20.0% | 8.0% | 8.0% | 28.0% | 8.0% | 14.0% | 14.0% | 14.0% | 2.0% | 6.0% | 16.0% | 16.0% |
| gpt-4o-2024-08-06 | 6.0% | 26.0% | 6.0% | 28.0% | 6.0% | 14.0% | 2.0% | 18.0% | 2.0% | 18.0% | 6.0% | 24.0% |
| claude-3.5-sonnet-20241022 | 66.0% | 48.0% | 24.0% | **58.0%** | 22.0% | **34.0%** | 16.0% | **58.0%** | 12.0% | **26.0%** | 26.0% | **60.0%** |
| Llama-3.1-405B-Instruct | 38.0% | 12.0% | 34.0% | 14.0% | 30.0% | 8.0% | 30.0% | 18.0% | 34.0% | 6.0% | 40.0% | 8.0% |
| Llama-3.3-70B-Instruct | 10.0% | 18.0% | 30.0% | 10.0% | 20.0% | 12.0% | 12.0% | 8.0% | 18.0% | 8.0% | 28.0% | 6.0% |
| DeepSeek-R1 | 70.0% | 14.0% | 48.0% | 20.0% | 36.0% | 20.0% | 44.0% | 30.0% | 34.0% | 12.0% | 58.0% | 38.0% |
| DeepSeek-V3 | 68.0% | 0.0% | 62.0% | 6.0% | **42.0%** | 10.0% | 48.0% | 4.0% | **42.0%** | 2.0% | **60.0%** | 18.0% |
| DeepSeek-Coder-V2.5-Instruct | 60.0% | 14.0% | 62.0% | 16.0% | 36.0% | 28.0% | 38.0% | 16.0% | 34.0% | 8.0% | 52.0% | 18.0% |
| Qwen-2.5-72B-Instruct | 22.0% | 6.0% | 2.0% | 8.0% | 0.0% | 8.0% | 0.0% | 4.0% | 0.0% | 6.0% | 2.0% | 6.0% |
| Qwen-2.5-Coder-32B-Instruct | 46.0% | 30.0% | 56.0% | 14.0% | 22.0% | 8.0% | 44.0% | 16.0% | 28.0% | 14.0% | 50.0% | 12.0% |
| **SF-Coder-32B-Instruct** | **76.0%** | **64.0%** | **78.0%** | 44.0% | 34.0% | 26.0% | **54.0%** | 38.0% | 34.0% | 22.0% | 50.0% | 42.0% |

| Model | gensim | | pydantic | | jinja | | transitions | | pylint | | pandas | |
|---|---|---|---|---|---|---|---|---|---|---|---|---|
| | replace | patch | replace | patch | replace | patch | replace | patch | replace | patch | replace | patch |
| o1-preview-2024-09-12 | 32.0% | 24.0% | 26.0% | 18.0% | 24.0% | 14.0% | 10.0% | 10.0% | 20.5% | 17.9% | 12.2% | 10.2% |
| o1-mini-2024-09-12 | 8.0% | 8.0% | 8.0% | 8.0% | 2.0% | 12.0% | 6.0% | 6.0% | 10.3% | 15.4% | 8.2% | 12.2% |
| gpt-4o-2024-08-06 | 4.0% | 16.0% | 6.0% | 26.0% | 2.0% | 18.0% | 14.0% | 6.0% | 2.6% | 15.4% | 0.0% | 10.2% |
| claude-3.5-sonnet-20241022 | 22.0% | **36.0%** | 20.0% | **42.0%** | 12.0% | **30.0%** | **26.0%** | 20.0% | 7.7% | **30.8%** | 14.3% | **28.6%** |
| Llama-3.1-405B-Instruct | 44.0% | 10.0% | 34.0% | 16.0% | 16.0% | 8.0% | 16.0% | 6.0% | 23.1% | 7.7% | 10.2% | 6.1% |
| Llama-3.3-70B-Instruct | 22.0% | 10.0% | 20.0% | 10.0% | 6.0% | 2.0% | 12.0% | 6.0% | 17.9% | 7.7% | 8.2% | 6.1% |
| DeepSeek-R1 | **50.0%** | 24.0% | 42.0% | 24.0% | 28.0% | 12.0% | 18.0% | 10.0% | 15.4% | 10.3% | **24.5%** | 8.2% |
| DeepSeek-V3 | 42.0% | 4.0% | 42.0% | 10.0% | **38.0%** | 2.0% | 14.0% | 6.0% | 17.9% | 10.3% | **24.5%** | 6.1% |
| DeepSeek-Coder-V2.5-Instruct | 40.0% | 14.0% | 38.0% | 18.0% | 32.0% | 10.0% | 14.0% | 6.0% | 23.1% | 20.5% | 22.4% | 10.2% |
| Qwen-2.5-72B-Instruct | 4.0% | 4.0% | 2.0% | 12.0% | 2.0% | 2.0% | 8.0% | 6.0% | 2.6% | 2.6% | 4.1% | 6.1% |
| Qwen-2.5-Coder-32B-Instruct | 38.0% | 12.0% | 42.0% | 14.0% | 26.0% | 10.0% | 4.0% | 4.0% | 15.4% | 10.3% | 16.3% | 8.2% |
| **SF-Coder-32B-Instruct** | **50.0%** | 34.0% | **50.0%** | 36.0% | **38.0%** | 24.0% | 20.0% | **28.0%** | **25.6%** | 17.9% | 16.3% | 14.3% |

During the evaluation of language models, we observed that the generated patches were almost entirely incompatible with system tools such as Linux's patch utility, making it impossible to apply them directly to the codebase. While the generated patches generally contain the correct modifications, they often fail to accurately define the contextual modification range. Due to this limitation, we apply a post-processing step, converting the generated patches into the **replace format**. We then use this replace-based approach to modify the codebase more effectively.

Table 5 presents the execution pass rates of 12 mainstream LLMs on **SWE-Flow-Bench (Lite)**. The results indicate that, with the exception of claude-3.5-sonnet, all other models perform poorly on this benchmark. Furthermore, claude-3.5-sonnet demonstrates significantly higher accuracy in the patch format compared to the replace format. This suggests that its training data likely contains a substantial amount of **patch format** software engineering data. In contrast, **SF-Coder-32B-Instruct**, which is fine-tuned on **SWE-Flow**-synthesized data, achieves significant performance improvements on these software engineering tasks. It consistently outperforms other models in both the replace and patch formats, reaching state-of-the-art performance levels.

## E.2. Evaluation Results of Agents

For the evaluation of agents, we employed the OpenHands (Wang et al., 2024a) framework with its default configuration, setting the maximum iteration limit to 30. Table 6 presents the performance of different language models integrated with OpenHands on **SWE-Flow-Bench (Lite)**. The results indicate that claude-3.5-sonnet consistently outperforms all other LLMs. However, even claude-3.5-sonnet struggles to successfully complete complex software engineering development tasks, as shown in the lower section of the table.

*Table 6.* Evaluation results of OpenHands with language models on **SWE-Flow-Bench (Lite)**.

| Model | arrow | | cryptography | | librosa | | marshmallow | | seaborn | | Pillow | |
|---|---|---|---|---|---|---|---|---|---|---|---|---|
| | Acc. | EV | Acc. | EV | Acc. | EV | Acc. | EV | Acc. | EV | Acc. | EV |
| claude-3.5-sonnet-20241022 | **80.0%** | **12.12** | **98.0%** | **15.29** | **72.0%** | **10.72** | **92.0%** | **13.95** | **82.0%** | **12.4** | **90.0%** | **14.34** |
| gpt-4o-2024-08-06 | 42.0% | 6.03 | 82.0% | 12.31 | 42.0% | 6.18 | 30.0% | 4.41 | 42.0% | 6.07 | 70.0% | 10.72 |
| deepseek-chat | 14.0% | 2.34 | 22.0% | 3.65 | 32.0% | 4.96 | 10.0% | 1.65 | 76.0% | 11.42 | 12.0% | 2.05 |
| Qwen2.5-Coder-32B-Instruct | 14.0% | 1.99 | 18.0% | 2.52 | 26.0% | 3.81 | 10.0% | 1.44 | 8.0% | 1.14 | 12.0% | 1.72 |
| Model | gensim | | pydantic | | jinja | | transitions | | pylint | | pandas | |
| | Acc. | EV | Acc. | EV | Acc. | EV | Acc. | EV | Acc. | EV | Acc. | EV |
| claude-3.5-sonnet-20241022 | **68.0%** | **10.37** | **50.0%** | **7.41** | **58.0%** | **8.51** | **8.0%** | **1.14** | **49.0%** | **7.52** | **48.0%** | **7.0** |
| gpt-4o-2024-08-06 | 30.0% | 4.4 | 12.0% | 1.73 | 28.0% | 4.1 | **8.0%** | 1.12 | 15.0% | 2.18 | 6.0% | 0.86 |
| deepseek-chat | 12.0% | 1.9 | 12.0% | 1.89 | 12.0% | 1.88 | 2.0% | 0.29 | 31.0% | 4.62 | 6.0% | 1.1 |
| Qwen2.5-Coder-32B-Instruct | 18.0% | 2.56 | 12.0% | 1.7 | 4.0% | 0.57 | 0.0% | 0.0 | 8.0% | 1.18 | 4.0% | 0.57 |

# F. Limitations and Future Work

## F.1. Limitations of sweflow

While **SWE-Flow** provides highly accurate and comprehensive function dependency analysis for synchronous programs, it has limitations in handling asynchronous execution and multi-process applications.

**Challenges in Asynchronous Program Analysis.** **SWE-Flow** is designed to track function dependencies in sequential execution flows. However, in asynchronous programs that rely on event loops, coroutine scheduling, and callback mechanisms (e.g., Python's asyncio), execution order is dynamic and non-deterministic. This makes it difficult to precisely reconstruct function call relationships, potentially leading to incomplete or imprecise dependency graphs in such cases.

**Incomplete Dependency Tracking in Multi-Process Applications.** In multi-process applications, function calls occur across independent memory spaces and communicate via inter-process communication (IPC) mechanisms such as message queues, shared memory, or sockets. Since **SWE-Flow** currently operates within a single process scope, it cannot fully capture function dependencies that span multiple processes, limiting its effectiveness for distributed execution analysis.

## F.2. Future Work

**Synthesizing More Challenging Data.** In this study, we employ the **SWE-Flow** framework to decompose a complex software engineering task into multiple simpler subtasks, each requiring only a minimal amount of incremental code development. Owing to the flexibility of **SWE-Flow**, we can also synthesize more challenging software engineering tasks. For instance, by merging multiple consecutive tasks within the development schedule generated by **SWE-Flow-Schedule** into a single new task and subsequently applying the same post-processing steps for data synthesis, we can construct a more demanding task. In the extreme case, where all development tasks are merged into a single task, this results in a scenario where the entire project must be developed from scratch.

**Enhancing Reinforcement Learning.** Recently, a surge of reasoning-aware models, such as o1 (Jaech et al., 2024), DeepSeek-R1 (Guo et al., 2025), and QwQ (Yang et al., 2024a), have emerged, all of which are trained using reinforcement learning. Reward feedback is a crucial component of reinforcement learning (Ouyang et al., 2022a); however, there remains a significant lack of large-scale, verifiable software engineering datasets. In contrast, the data synthesized by our **SWE-Flow** framework inherently possesses verifiability, as the correctness of model-generated content can be directly assessed by executing the generated code within the corresponding containerized environment. We believe that the **SWE-Flow** framework will significantly facilitate the future reinforcement learning training of code generation models.

**Enhancing Pre-training.** The pretraining dataset directly determines the fundamental capabilities of a large language model (LLM). However, due to the scarcity of verifiable software engineering data, current open-source LLMs suffer from a severe lack of high-quality software engineering data in their pre-training corpora. By leveraging a vast number of GitHub projects with **continuous integration (CI)** enabled environments, we can already synthesize an extensive amount of verifiable training data. Furthermore, ongoing research (Zhang et al., 2025) is exploring automated dependency environment construction for software projects, which will further expand the scale of data that can be synthesized using the **SWE-Flow** framework. We believe that incorporating a large volume of synthetic, verifiable software engineering data into the pre-training corpus of open-source models will significantly enhance the foundational capabilities of LLMs in the domain of code generation and software development.

## G. Runtime Dependency Graph

Figures 10, 11, 12, and 13 illustrate examples of Runtime Dependency Graphs (RDGs) constructed by **SWE-Flow** from the Hugging Face Datasets library.

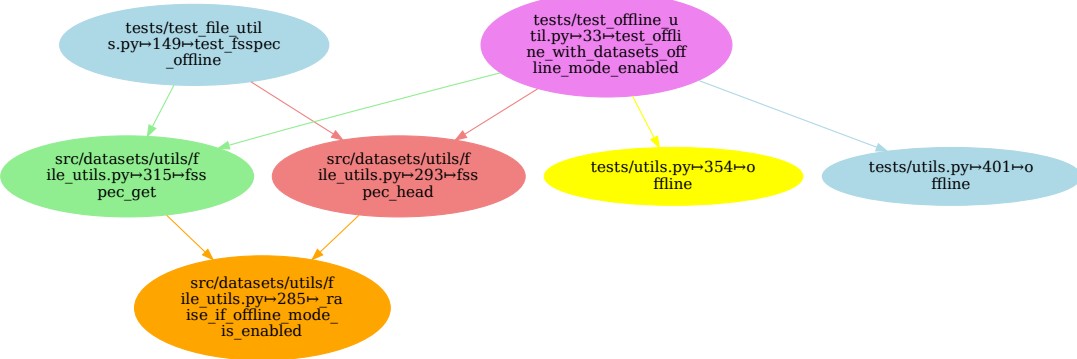

*Figure 10.* An Runtime Dependency Graph (RDG) instance form *datasets* project.

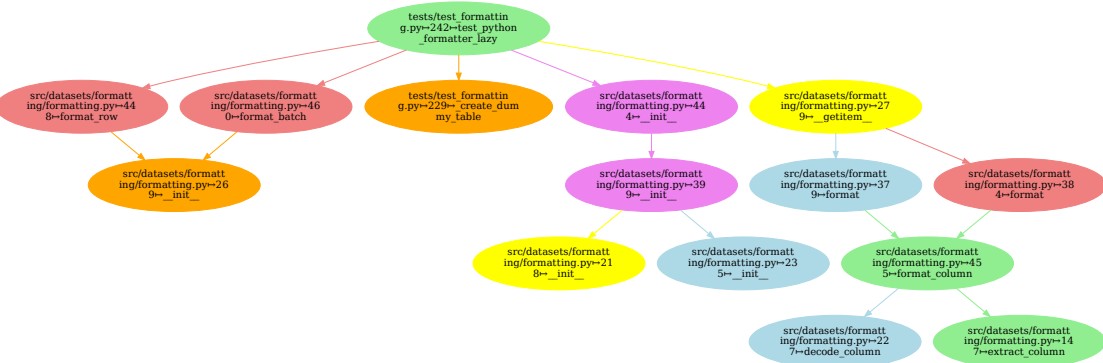

*Figure 11.* An Runtime Dependency Graph (RDG) instance form *datasets* project.

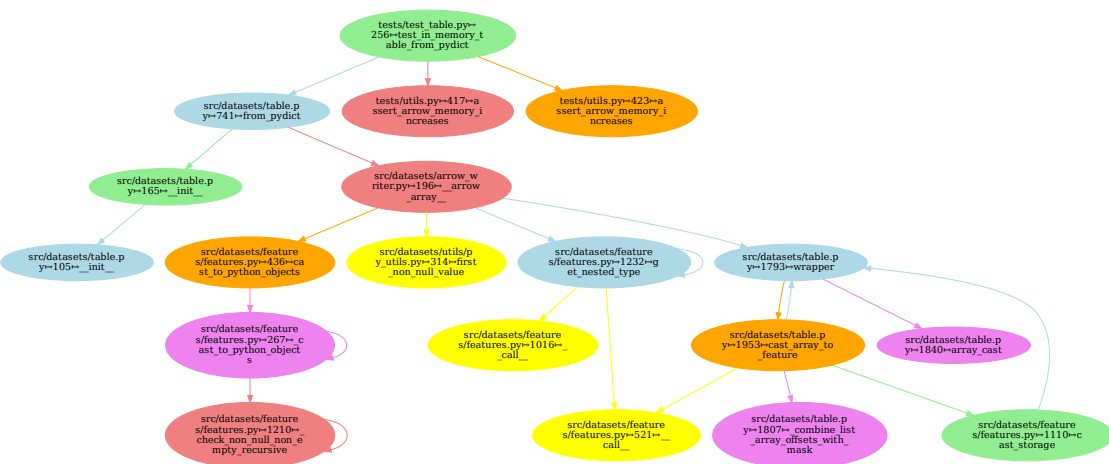

*Figure 12.* An Runtime Dependency Graph (RDG) instance form *datasets* project.

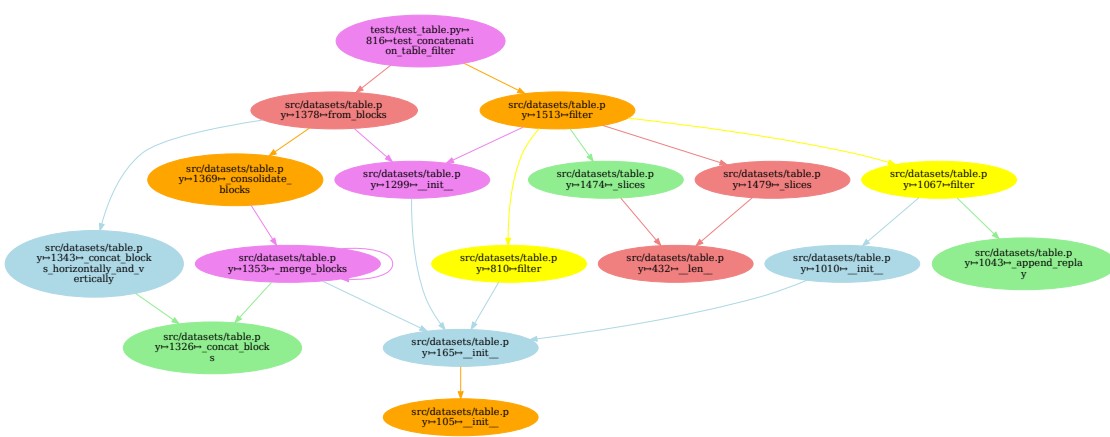

*Figure 13.* An Runtime Dependency Graph (RDG) instance form *datasets* project.

