# OpenReview forum: "Synthesizing Software Engineering Data in a Test-Driven Manner"
_ICML.cc/2025/Conference — ICML 2025 poster_

### Official Review · Reviewer_Ep2m · 2025-03-11

**Overall Recommendation:** 3

**Summary:**

This paper proposes a TDD-driven data synthesis framework `UnitFlow`, which can
generate data samples for incremental development tasks based on real-world
GitHub project and unit tests. Based on this framework, this paper constructs a
promising benchmark `UnitFlow-Eval`, which contains data samples from 74
real-world Python projects. Fine-tuning code models on this dataset can
effectively enhance LLM performance in incremental development tasks. This paper
has practical value for the development of `AI for SE` and code models but still
has issues to be clarified.

## update after rebuttal

I have updated my review after the rebuttal. Main concerns are addressed so I raised the score.

**Claims And Evidence:**

The authors provide the necessary preliminaries and definitions in Section 2 and
exhibit detailed experimental results and benchmark information in the Appendix.

However, the authors fail to provide sufficient evidence to support the claimed
contribution, namely, `effectively enhances LLM performance in incremental
development tasks'. Specifically, the author only shows the improvement of the
Pass Rate after finetuning one single model on their dataset. It will better
illustrate the practical significance and contribution of this benchmark if the
authors can provide some comparisons of the fine-tuning results of other models
(even small models of 7b and 14b) or show the performance of the fine-tuned LLM
on other development tasks/datasets.

**Essential References Not Discussed:**

N/A

**Experimental Designs Or Analyses:**

The experiment cannot provide sufficient evidence for the claimed contributions.

Only involving 74 projects in the Python programming language, limits the practical value of the benchmark.

**Methods And Evaluation Criteria:**

The paper itself proposes to generate a benchmark, which I think the synthesis method is promising and should be studied. It is still hard to demonstrate that the generated data aligns with real-world data.

User studies were not conducted to get a sense of the quality and practicality of the generated dataset.

**Other Comments Or Suggestions:**

The axes in Figure 5 are not suitable. Its caption claims that each axis is a software engineering task, but it actually refers to a repository. Such a design seems to follow the design of the SWE benchmark (Figure 4 in their paper).

However, UnitFlow-Eval and SWEbenchmark are different, and this paper breaks up a project into several incremental development tasks. In this case, simply evaluating the effectiveness of LLM code generation on a project is not appropriate. I think a more appropriate approach might be to summarize the types of tasks involved in incremental development. For example, some incremental development solves bugs in the code, while some supplement new functions. They are different tasks with different targets.

The authors may consider summarizing these incremental development tasks and then designing new axes (each one is an incremental development task). Based on such classification, the evaluation of the pass rate and efficiency value of different LLMs will be more in line with the needs of incremental development scenarios. New axes can also allow this benchmark to measure the capabilities of LLM in incremental development.

**Other Strengths And Weaknesses:**

Cannot find `Figure 5 in appendix` mentioned in the right part of line 319.  Figure 5 is on page 8 and shows the efficiency.

**Questions For Authors:**

This paper uses LLMs to generate development documents and docstring in Lines 212 and 233. I wonder how authors deal with the hallucinations in the LLM generation.

**Relation To Broader Scientific Literature:**

This paper proposes a data synthesis framework, which is guided by the concept
of TDD to synthesize code samples for incremental development tasks from
open-source repositories. It is related to the prior SWE-benchmark but is
different. This paper splits one project into incremental steps, in line with
the process of real-world software development. This paper has cited appropriate
related works in Section 7 and Appendix and discussed differences between this
work and prior works.

**Theoretical Claims:**

n/a

---

> ### Author Rebuttal · Authors · 2025-03-31
>
> Dear reviewer,
>
> We are grateful for the reviewer’s valuable suggestions. Our detailed responses to the concerns are provided below.
>
> ---
> > **Concern D1:** The paper would benefit from additional empirical evidence to substantiate the claimed improvements in LLM performance on incremental development tasks, such as comparisons with other fine-tuned models or evaluations on other benchmarks.
>
> We fine-tuned **Qwen2.5-Coder-32B-Instruct** using the **UnitFlow synthetic dataset** and evaluated its performance on **SWE-Bench-Verified** using the **Agentless** framework. The results are as follows, reported as the average over three runs with standard deviation:
>
> | **Model** | **SWE-Bench-Verified Accuracy** |
> | - | :-: |
> | Qwen2.5-Coder-32B-Instruct | 33.79% ± 0.32% |
> | +Ours (fine-tuned with UnitFlow data) | **35.27%** ± 0.26% |
>
> **UnitFlow-Bench** evaluates models on incremental software engineering tasks, whereas **SWE-Bench-Verified** targets automated bug fixing. Improvements observed on both benchmarks demonstrate the robustness and effectiveness of our synthetic data generation approach.
>
> > **Concern D2:** It is still hard to demonstrate that the generated data aligns with real-world data.
>
> UnitFlow is explicitly designed to simulate real-world incremental software development, based on a detailed analysis of unit tests from real open-source projects.
>
> Specifically, we:
> - Use existing unit tests to define realistic development goals,
> - Synthesize code changes that mirror typical developer behavior,
> - Ensure the process is traceable, reproducible, and guided by real testing practices.
>
> All instances are validated in executable environments, ensuring both correctness and practical feasibility. In summary, the design, source, and validation process of UnitFlow strongly support its alignment with real-world development workflows.
>
> > **Concern D3:** User studies were not conducted to get a sense of the quality and practicality of the generated dataset.
>
> While user studies can be valuable for assessing perceived quality, our focus is on systematic, executable validation: each UnitFlow-generated instance is tested in a execution verifiable environment to ensure that:
> - the code runs correctly, and
> - the intended behavior is preserved or modified as expected.
>
> This process provides a scalable, objective, and reproducible measure of data quality, which we believe is more precise and reliable than subjective user feedback, especially for software engineering tasks
>
> > **Concern D4:** The experiment cannot provide sufficient evidence for the claimed contributions. Only involving 74 projects in the Python programming language, limits the practical value of the benchmark.
>
> We would like to clarify that the current scale of our dataset has significantly expanded.
>
> Since the initial submission, we have successfully constructed over 6,000 executable environments across diverse real-world Python projects. Based on these, we have synthesized over 200k validated data instances, making UnitFlow-Bench a substantially larger and more practical benchmark than initially reported.
>
> Moreover, models fine-tuned on UnitFlow-synthesized data demonstrate consistent performance gains on UnitFlow-Bench-Lite and SWE-Bench-Verified. These results highlight the effectiveness and generalizability of UnitFlow-generated data, further supporting the practical value and impact of the benchmark.
>
> To support further research and demonstrate practical scalability, **we will release all synthetic data and executable environments**.
>
> > **Concern D5:** This paper uses LLMs to generate development documents and docstring in Lines 212 and 233. I wonder how authors deal with the hallucinations in the LLM generation.
>
> As described in Section 3.4, we manually inspected samples from several models before scaling up generation. For tasks such as generating development instructions and docstrings, Qwen2.5-Coder-32B-Instruct performed consistently well, and we did not observe hallucinations in the sampled outputs, they were semantically aligned with the code or tests.
>
> Additionally, all generated instances undergo executable validation: if the synthetic data fail runtime checks, they are automatically discarded. This ensures that only accurate and functionally correct data are retained, effectively filtering out hallucinated or inconsistent outputs.
>
> > **Concern D6:** Figure 5 not found in appendix.
>
> The prompt format figure referenced in line 319 was accidentally left out during compilation. We will make sure it is included in the revised manuscript.
>
> ---
> We’re happy to clarify any remaining questions you may have. If you feel our response resolves your concerns, we would be grateful if you could consider raising your score.

---

### Official Review · Reviewer_ffV1 · 2025-03-12

**Overall Recommendation:** 3

**Summary:**

This paper introduces UnitFlow, a novel data synthesis framework based on Test-Driven Development (TDD), which automatically generates software engineering data by inferring incremental development steps directly from unit tests. The framework constructs a Runtime Dependency Graph (RDG) to capture function interactions, enabling the generation of structured development schedules and verifiable TDD tasks. In addition, the authors created the UnitFlow-Eval benchmark by generating 16,061 training instances and 2,020 test instances from real-world GitHub projects. The paper demonstrates that fine-tuning a large language model (LLM) on this synthetic dataset significantly enhances performance in TDD-based programming.

**Claims And Evidence:**

Some claims are not supported by clear and convincing evidence. The paper could show/prove the validity and quality of the synthesized software engineering data. Is the synthesized data valid and of high quality?

**Essential References Not Discussed:**

The paper did not consider the scenario of software evolution, where software (including its structure/dependency graph) keeps changing, such as those discussed in the following paper:
Dewu Zheng et al., Towards more realistic evaluation of LLM-based code generation: an experimental study and beyond
https://arxiv.org/pdf/2406.06918.

**Experimental Designs Or Analyses:**

The model UF-Coder-32B-Instruct is actually a fine-tuned version via synthesized training data, so I am not sure if the performance improvement on UnitFlow-Bench-Lite can demonstrate the effectiveness of UnitFlow-synthesized data.

**Methods And Evaluation Criteria:**

Pass Rate and Efficiency Value are used to measure the performance of several different LLM on UnitFlow-Bench-Lite. The paper shows that the limitations of current large language models in handling practical software engineering challenges could be addressed by fine-tuning LLMs with the synthesized data. However, the authors only fine-tuned the Qwen2.5-Coder-32B-Instruct model and evaluated it on the UnitFlow-Bench-Lite test set. More experiments could be conducted.

**Other Comments Or Suggestions:**

. Figure 3 never get referred in the whole paper.

. Page 8, line 398. “The evaluation results are presented in Figure 5 and Table 6 in the appendix”, Figure 5 is actually not in the appendix.

. Page 6, line 308 and line 318. Line 308 “Finally, we obtain 74 projects that can pass all the unit tests in the installed test environment.” Line 318 “We only keep the unit tests that pass.” Line 318 implies that there are unit tests that cannot pass while line 308 claims that all unit tests can pass. Are there any differences between the criteria in the ‘Preparation of Test Environment’ and ‘Verifiable Data Generation’?

. “Among these projects, 12 projects are selected for testing, and the remaining 62 projects are used for training”. Did you select these projects for testing and training randomly or not? Any criteria for projects selection?

. Page 3, line 3 “UnitFlow removes the implementation of core functions covered by the current…” and Page 4, line 205 “that cover the same Core Function Nodes (CFNs).” Does these “Core Functions” refer to both Target Core Function and Dependent Core Function?

. Page 7, line 317, “The pass rate is defined as the ratio of successfully completed tasks (those that pass unit tests)…” The word "task" is confusing. Does it mean an unit test? What is the granularity of a task?

. Page 4, line 205 “we first merge all TTFNs that cover the same Core Function Nodes (CFNs)”. Does the CFNs refer to both Target Core Function Node and Dependent Core Function Node? Any clarification?

**Other Strengths And Weaknesses:**

The paper is generally well written.

The data and code are not publicly available (data link cannot be found in the paper)

**Questions For Authors:**

As the model UF-Coder-32B-Instruct is actually fine-tuned using synthesized training data, can the performance improvement on UnitFlow-Bench-Lite demonstrate the effectiveness of UnitFlow-synthesized data?

**Relation To Broader Scientific Literature:**

N/A

**Theoretical Claims:**

The paper does not contain theoretical proofs and claims.

---

> ### Author Rebuttal · Authors · 2025-03-31
>
> Dear reviewer,
>
> We sincerely appreciate your thoughtful and constructive feedback. Please find our responses to the concerns below.
>
> ---
>
> > **Concern C1:** Is the synthesized data valid and of high quality?
>
> We fine-tuned the **Qwen2.5-Coder-32B-Instruct** model using data synthesized by UnitFlow, resulting in a substantial accuracy improvement on the UnitFlow-Bench benchmark. To further validate the effectiveness of UnitFlow-generated data, we conducted evaluations using the **Agentless** framework on the **SWE-Bench-Verified** dataset, reporting results as the average over three runs with standard deviation.
>
> | **Model** | **SWE-Bench-Verified Accuracy** |
> | - | :-: |
> | Qwen2.5-Coder-32B-Instruct | 33.79% ± 0.32% |
> | +Ours (fine-tuned with UnitFlow data) | **35.27%** ± 0.26% |
>
> The consistent improvements observed on both **UnitFlow-Bench** and **SWE-Bench-Verified** demonstrate the practical utility and generalizability of the synthetic data produced by UnitFlow.
>
> > **Concern C2:** The model UF-Coder-32B-Instruct is actually a fine-tuned version via synthesized training data, so I am not sure if the performance improvement on UnitFlow-Bench-Lite can demonstrate the effectiveness of UnitFlow-synthesized data.
>
> As mentioned above, we also fine-tuned **Qwen2.5-Coder-32B-Instruct** and evaluated it on **SWE-Bench-Verified**. The results provide further evidence of the effectiveness of the synthetic data generated by **UnitFlow**.
>
> > **Concern C3:** The paper did not consider the scenario of software evolution, where software keeps changing.
>
> We are exploring ways to incorporate software evolution into our setting, but we have found it to be a challenging problem.
>
> We constructed executable and verifiable environments for 1,276 repositories related to SWE tasks. However, since the environments were built based on the latest commit on the main branch, some environments became invalid when checking out historical commits—due to changes in the software’s structure or dependencies over time. As a result, out of the 24,961 instances covered by these repositories, only 1,242 instances could be successfully validated using the prebuilt environments.
>
> > **Concern C4:** Are there any differences between the criteria in the ‘Preparation of Test Environment and Verifiable Data Generation’?
>
> During the construction of executable verification environments, we use the **exit code of pytest** as an indicator of environment status. Some key exit codes are interpreted as follows:
>
> - **Exit code 0**: pytest ran successfully and all unit tests passed.
> - **Exit code 1**: pytest ran successfully but some unit tests failed.
> - **Exit code 3**: pytest failed to run, typically due to import errors caused by an incomplete or broken environment.
>
> In the environment construction phase, we treat both **exit code 0 and 1** as indications of a **successfully built environment**. However, during the data synthesis phase, we retain **only instances where unit tests pass** (exit code 0), to ensure that each synthesized data point can be **verified within the current environment**. Instances that fail unit tests are discarded to maintain data quality.
>
> > **Concern C5:** “Among these projects, 12 projects are selected for testing, and the remaining 62 projects are used for training”. Did you select these projects for testing and training randomly or not? Any criteria for projects selection?
>
> The **test projects** were selected based on **functional diversity**. Specifically, we chose 12 projects that serve **distinct purposes** (e.g., visiualization, data processing) to ensure broader coverage. The selection criteria also included **popularity (GitHub star count)** and **recent activity** (i.e., whether there were commits within the past six months).
>
> We will clarify the project selection criteria more explicitly in the revised paper.
>
> > **Concern C6:** Does these “Core Functions” refer to both Target Core Function and Dependent Core Function?
>
> Yes, your understanding is correct. We will clarify this in the revised version of the paper.
>
> > **Concern C7:** The word "task" is confusing. Does it mean an unit test? What is the granularity of a task?
>
> Yes, by "task" here, we are referring to whether the given development task can be completed successfully and pass the corresponding unit tests.
>
> > **Concern C8:** Does the CFNs refer to both Target Core Function Node and Dependent Core Function Node? Any clarification?
>
> Yes, your understanding is correct. We will clarify this in the revised version of the paper.
>
> > **Concern C9:** Typos
>
> We sincerely appreciate your careful reading and for identifying the typos in **Figure 3** and **Figure 5**. These will be fixed in the revised version.
>
> ---
> Please feel free to reach out. We would be happy to clarify any remaining questions you may have. If you find our response addresses your concerns, we sincerely hope you would consider raising your score accordingly.

---

### Official Review · Reviewer_YdKw · 2025-03-14

**Overall Recommendation:** 3

**Summary:**

This paper proposes a data synthesis framework UnitFlow that leverages Test-Driven Development to automatically generate high-quality, structured, and verifiable training data for LLMs in software engineering. It constructs a Runtime Dependency Graph (RDG) from unit tests to capture function interactions and generates a step-by-step development schedule. For each step, UnitFlow produces a partial codebase, a requirement document based on unit tests, and a reference solution. Using UnitFlow, the authors synthesized a dataset of 16k training instances and 2k test instances, which they used to fine-tune the Qwen2.5-Coder-32B-Instruct model, resulting in the UF-Coder-32B-Instruct model. Experiments on the UnitFlow-Eval benchmark demonstrated significant improvements in the model's ability to perform TDD-based coding tasks.

## update after rebuttal
I appreciate the rebuttal by authors. My concern has been addressed and I would like to keep my score.

**Claims And Evidence:**

Yes.

**Essential References Not Discussed:**

No.

**Experimental Designs Or Analyses:**

The evaluation of the proposed method is limited to the UnitFlow-Eval benchmark, which shares the same distribution as the training data. The authors did not conduct experiments on other software development benchmarks such as SWE-Bench. This limitation may affect the assessment of the method's generalizability across different software development contexts and tasks.

**Methods And Evaluation Criteria:**

Yes, but could be further improved.

**Other Comments Or Suggestions:**

No.

**Other Strengths And Weaknesses:**

Other Strengths:
- Clear writing and organization.
- Scalablility of the proposed method.

Other Weaknesses:
- While the approach demonstrates theoretical scalability, the volume of synthetic data and the experimental evaluation are confined to supervised fine-tuning.

**Questions For Authors:**

Please refer to Experimental Designs and above Weaknesses.

**Relation To Broader Scientific Literature:**

No.

**Theoretical Claims:**

Not applicable.

---

> ### Author Rebuttal · Authors · 2025-03-31
>
> Dear reviewer,
>
> We are grateful for your valuable suggestions. Our detailed responses to the concerns are provided below.
>
> ---
>
> > **Concern B1:** The evaluation of the proposed method is limited to the UnitFlow-Eval benchmark, which shares the same distribution as the training data. The authors did not conduct experiments on other software development benchmarks such as SWE-Bench. This limitation may affect the assessment of the method's generalizability across different software development contexts and tasks.
>
> We agree that evaluating the generalizability of the proposed method beyond the UnitFlow-Eval benchmark is important. In addition to our main evaluation on UnitFlow-Eval, we conducted further experiments using the **SWE-Bench-Verified** benchmark, which differs significantly from our training data in both distribution and task format. Specifically, we fine-tuned **Qwen2.5-Coder-32B-Instruct** with UnitFlow synthetic data, and evaluated it using the **Agentless** framework. The results are as follows, reported as the average over three runs with standard deviation:
>
> | **Model** | **SWE-Bench-Verified Accuracy** |
> | - | :-: |
> | Qwen2.5-Coder-32B-Instruct | 33.79% ± 0.32% |
> | +Ours (fine-tuned with UnitFlow data) | **35.27%** ± 0.26% |
>
> This improvement of **+1.48%** indicates that our method not only improves performance on in-distribution tasks but also enhances the model’s ability to generalize to real-world software engineering tasks under different settings.
>
> We appreciate the reviewer’s suggestion and will include these results in the revised version to strengthen the generalizability evaluation of our method.
>
>
> > **Concern B2:** While the approach demonstrates theoretical scalability, the volume of synthetic data and the experimental evaluation are confined to supervised fine-tuning.
>
> Since the initial submission, we have significantly scaled up the synthetic data generation process. Specifically, we have automated the construction of 6,008 executable repository images and used *UnitFlow* to synthesize over **200k** task instances.
>
> Our analysis suggests that supervised fine-tuning (SFT) benefits saturate  around **20k–30k** instances. Therefore, we plan to leverage the additional data in the **continued pretraining** phase, which contributes to improving general code understanding and task generalization.
>
> To support further research and demonstrate practical scalability, **we will release all synthesized data and executable environments**.
>
> ---
> Please feel free to reach out if you have any further questions — we’d be happy to clarify. If you find our response addresses your concerns, we sincerely hope you would consider raising your score accordingly.

---

### Official Review · Reviewer_qTpi · 2025-03-15

**Overall Recommendation:** 4

**Summary:**

The paper introduces UnitFlow, a novel framework for synthesizing test-driven software engineering data. Unlike prior datasets that rely on human-submitted issues, UnitFlow generates incremental development steps directly from unit tests. The framework constructs a Runtime Dependency Graph (RDG) to capture function interactions, enabling structured development tasks that mirror real-world iterative coding practices. Evaluation of state-of-the-art LLMs and agents, demonstrating that current models struggle with complex software engineering workflows.

**Claims And Evidence:**

The core claims are well-supported by experimental results.

Potential Limitations:

- Comparison with other benchmarks is limited. The paper discusses SWE-Bench but does not provide a direct side-by-side comparison in terms of realism or task difficulty.
- The robustness of UnitFlow-generated data against real-world code review practices is unclear.

**Essential References Not Discussed:**

No.

**Experimental Designs Or Analyses:**

++ The choice of models is reasonable, covering OpenAI, Google, and specialized models.
++ The benchmarking methodology is sound.
-- The paper lacks statistical significance tests for performance differences—are observed differences robust under variations in prompts or datasets?

**Methods And Evaluation Criteria:**

++ The Runtime Dependency Graph (RDG) effectively captures function dependencies.

++ Using unit test execution traces instead of static analysis improves accuracy in identifying function interactions.

++ Benchmarking both LLMs and agents provides a systematic and well-defined evaluation.

-- The evaluation does not compare against human-written commit sequences, which could serve as a valuable real-world baseline.

-- The study does not report how often models generate nearly correct but slightly misformatted code, which may impact pass rates.

**Other Comments Or Suggestions:**

- Section 4.1, line 319: Reference to Figure 5 appears incorrect.
- Section 5.2, line 370: The term "unitflow-bench-lite" should be introduced earlier for clarity.

**Other Strengths And Weaknesses:**

Strengths:

- Automated and scalable dataset generation.
- Benchmark evaluates LLMs in an iterative development setting, making it more realistic than static function-based tasks.
- Fine-tuned LLM outperforms baselines, validating the effectiveness of the dataset.

Weaknesses:

- No qualitative analysis of whether models learn good coding practices or just pattern-match training data.

**Questions For Authors:**

1. Have you analyzed how often models generate near-correct solutions with minor formatting issues (e.g., missing imports, minor syntax mistakes)?
2. How does UnitFlow compare to human-written commit sequences in terms of task structure and complexity?

**Relation To Broader Scientific Literature:**

- Prior benchmarks like HumanEval, MBPP, and SWE-Bench focus on single-step or static QA, whereas UnitFlow emphasizes incremental development.
- Unlike SWE-Bench, which extracts tasks from GitHub issues, UnitFlow synthesizes development tasks, making it more scalable and controllable.

**Theoretical Claims:**

Not applicable (no formal proofs or theoretical claims in the paper).

---

> ### Author Rebuttal · Authors · 2025-03-31
>
> Dear reviewer,
>
> We deeply appreciate your thoughtful review and your recognition of our contributions. Below, we provide point-by-point responses to your concerns and suggestions.
>
> ---
>
> > **Concern A1:** The evaluation does not compare against human-written commit sequences, which could serve as a valuable real-world baseline.
>
> Given that the unitflow-bench benchmark comprises more than 2,000 test cases, conducting experiments with human-written commits is quite time-intensive. Regrettably, we were unable to complete these experiments within the rebuttal timeframe. Nonetheless, we are committed to performing the experiments and will include the results in the revised version of our paper.
>
>
> > **Concern A2:** The study does not report how often models generate nearly correct but slightly misformatted code, which may impact pass rates.
>
> We calculated the accuracy of the solution format generated by each model and present the results in the table below.
>
> |Model|Empty Patch|Empty Replace|
> |-|:-:|:-:|
> |claude3.5-sonnet-1022|0.00%|0.00%|
> |DeepSeek-Coder-V2-Instruct-0724|2.00%|1.67%|
> |DeepSeek-R1|1.67%|1.67%|
> |DeepSeek-V3|4.01%|6.66%|
> |gpt-4o-2024-08-06|0.00%|0.00%|
> |Llama-3.1-405B-Instruct|6.67%|8.03%|
> |Llama-3.3-70B-Instruct|2.04%|1.04%|
> |o1-mini-2024-09-12|5.34%|3.33%|
> |Qwen2.5-72B-Instruct|3.51%|0.59%|
> |Qwen2.5-Coder-32B-Instruct|2.71%|0.33%|
> |UF-Coder-32B-Instruct|4.17%|1.38%|
>
>
> > **Concern A3:** The paper lacks statistical significance tests for performance differences, are observed differences robust under variations in prompts or datasets?
>
> In the evaluation, we standardized only the format of the prompts, which contained solely the essential formatting instructions. Since the actual content of the prompts changed significantly based on the contextual information of each test instance, the robustness of the experimental results is preserved.
>
>
> > **Concern A4:** No qualitative analysis of whether models learn good coding practices or just pattern-match training data.
>
> Our UF-Coder-32B-Instruct model is trained based on Qwen2.5-Coder-32B-Instruct. As shown in the error format statistics table above, the proportion of formatting errors generated by UF-Coder-32B-Instruct is comparable to that of Qwen2.5-Coder-32B-Instruct — in fact, the latter even exhibits a slightly lower error rate. However, the accuracy of UF-Coder-32B-Instruct is significantly higher than that of Qwen2.5-Coder-32B-Instruct. This suggests that UF-Coder-32B-Instruct has not merely memorized patterns from the training data, but has indeed improved its capability in software development tasks.
>
>
> > **Concern A5:** How does UnitFlow compare to human-written commit sequences in terms of task structure and complexity?
>
> Compared to human-written commit sequences, which often vary in granularity and mix different types of changes, UnitFlow provides a more structured and goal-driven development trajectory. By aligning each commit with the satisfaction of a specific unit test, UnitFlow enforces a clear functional decomposition and promotes modular, incremental progress. Despite being automatically generated, the commit sequences can capture meaningful task dependencies and span multiple components, reflecting a level of complexity comparable to real-world development. This makes UnitFlow a valuable tool for generating realistic and reproducible development workflows, particularly in settings where systematic task structure and functional clarity are desired.
>
>
> > **Concern A6:** Typos and Suggestions
>
> We will incorporate these improvements in the revised version.
>
> ---
> We sincerely appreciate your thoughtful and positive review. Should you have any further questions or suggestions, we would be glad to provide additional clarification.

---

### Decision · Program_Chairs · 2025-05-01

**Decision:**

Accept (poster)

**Comment:**

**Meta-Review of “UnitFlow: Synthesizing Software Engineering Data in a Test-Driven Manner”**

**Summary of the Paper**
This paper introduces **UnitFlow**, a novel framework designed to synthesize step-by-step software development tasks based on test-driven development (TDD). The approach departs from previous task creation approaches that rely on human-submitted issues or single-function tasks. Instead, UnitFlow automatically constructs a **Runtime Dependency Graph (RDG)** by running unit tests and capturing function interactions, then “skeletonizes” partial code bases to reflect incremental development steps. From these steps, the authors create the **UnitFlow-Eval** benchmark—over 16,000 training instances and 2,000 test instances—aimed at measuring and improving LLM performance in TDD-style software engineering. The experiments show that fine-tuning an open-source large language model (Qwen2.5-Coder-32B-Instruct) on the synthesized dataset yields notable gains in pass rates for test-driven development tasks.

---

## Strengths

1. **TDD-Based Data Synthesis**
   Reviewers generally agreed that using unit tests to guide data generation is a worthwhile idea. By using *executed* tests as the driver for incremental code construction, the paper sidesteps the noise and variability of human commits and issue reports, while still maintaining verifiability and clarity on intended functionality.

2. **Executable, Verifiable Tasks**
   Each synthetic task is validated in a containerized environment, ensuring correctness checks through real unit tests. This explicit requirement that tasks pass or fail in a fully functional environment strengthens the benchmark.

3. **Demonstrated Performance Gains**
   Fine-tuning a code LLM on the UnitFlow dataset results in improved performance on the in-domain **UnitFlow-Eval** task. Authors rebuttals show small accuracy improvement in the out-of-domain **SWE-Bench-Verified** task as well. This indicates that UnitFlow data may transfer beyond its own test distribution, possibly addressing a common reviewer request about generalizability.

4. **Scalable Data Generation**
   The authors show that, after the initial submission, they were able to scale up UnitFlow to cover over 6,000 Python repositories, synthesizing more than 200,000 validated tasks. This indicates the framework’s potential to produce large-scale training corpora for code-centric LLMs.

---

## Weaknesses and Limitations

1. **Clarity on Node Removal and Typographical Issues**
   The initial manuscript confused TTFNs (test-related nodes) and TCFNs (core function nodes) in a few places, causing reviewers and the area chair to question exactly which functions are preserved or stripped from the code base. The authors clarified that only core function implementations (TCFNs, DCFNs) are removed, while test functions remain intact. A revision is needed to fix the naming/terminology errors and ensure consistency between Figure 1 and the text.

2. **Experimental Scope and Real-World Comparison**
   Multiple reviewers noted the limited experiments:
   - The paper primarily focuses on **UnitFlow-Eval** which shares the same distribution as the training data. There are limited side-by-side comparisons. The rebuttal added data on SWE-Bench-Verified.
   - No direct human-commit or human-developer “baselines” are measured.
   While the rebuttal mitigates this somewhat by testing on SWE-Bench-Verified, additional data points or real commit sequences would strengthen claims about real-world applicability.

3. **Focus on Python Projects**
   Although the authors mention ongoing scaling efforts, the current benchmark remains Python-centric (74 initial repositories in the main experiments, or 6,000 in their scaled version). Cross-language generalization and broader coverage would be needed to show approach’s applicability to other languages.

4. **Incremental Development Complexity**
   Several reviewers suggested that real-world incremental coding can be more convoluted than the TDD-inspired steps extracted by UnitFlow. For instance, code can shift across directories, library versions can break older commits, and developer commit habits can be ad hoc. While the authors describe “skeletonization” plus function removal, the real-world unpredictability is missing. Addressing genuine software evolution scenarios remains a future challenge.

5. **Limited Agent Evaluations**
   Though the paper shows how UnitFlow-Eval can be used with agents, only OpenHands is tested. More comprehensive agent evaluations might be necessary.

---

## Recommendation

All four reviewers converged on acceptance or borderline-acceptance.
I recommend acceptance.